# CREMA: Generalizable and Efficient Video-Language Reasoning via Multimodal Modular Fusion

**Shoubin Yu**[*]   **Jaehong Yoon**[*]   **Mohit Bansal**
UNC Chapel Hill
{shoubin, jhyoon, mbansal}@cs.unc.edu

## Abstract

Despite impressive advancements in recent multimodal reasoning approaches, they are still limited in flexibility and efficiency, as these models typically process only a few fixed modality inputs and require updates to numerous parameters. This paper tackles these critical challenges and proposes CREMA, a generalizable, highly efficient, and modular modality-fusion framework that can incorporate many new modalities to enhance video reasoning. We first augment multiple informative modalities (such as *optical flow*, *3D point cloud*, *audio*, *thermal heatmap*, and *touch map*) from given videos without extra human annotation by leveraging sensors or existing pre-trained models. Next, we introduce a query transformer with multiple parameter-efficient modules associated with each accessible modality. It projects diverse modality features to the LLM token embedding space, allowing the model to integrate different data types for response generation. Furthermore, we propose a novel progressive multimodal fusion design supported by a lightweight fusion module and modality-sequential training strategy. It helps compress information across various assisting modalities, maintaining computational efficiency in the LLM while improving performance. We validate our method on 7 video-language reasoning tasks assisted by diverse modalities, including conventional VideoQA and Video-Audio/3D/Touch/Thermal QA, and achieve better/equivalent performance against strong multimodal LLMs, including OneLLM, BLIP-2, and SeViLA while reducing over 90% trainable parameters. We provide extensive analyses of CREMA, including the impact of each modality on reasoning domains, the design of the fusion module, and example visualizations. [1]

## 1 Introduction

We humans understand the world through various senses, such as sight, sound, touch, and heat, allowing us to understand our environment and act accordingly. This concept has inspired the field of multimodal learning that connects various perceptions, including vision-language (Alayrac et al., 2022; Li et al., 2023b; Zang et al., 2023; Radford et al., 2021), audio-video (Han et al., 2020; Tang et al., 2022), and 2D-3D joint vision (Li et al., 2020; Hou et al., 2021; 2023; Lei et al., 2024). In particular, recent Multimodal Large Language Models (MLLMs) (Yu et al., 2023b; Li et al., 2023b; Liu et al., 2023a; Tang et al., 2023a) have shown promising versatility in handling multiple forms of input data, such as vision, audio, and text. These models are crucial in real-world applications that require a comprehensive understanding of multiple modalities to make decisions in various contexts. For example, autonomous vehicles rely on visual road signs, sirens, and LIDAR for navigation and safe driving. Embodied AI takes visual, heat, and touch information to complete household tasks. Similarly, educational AI enhances the learning experience by integrating various information, such as videos, speech, and textbooks.

Despite their recent advancements, deploying a generic MLLM that handles multiple diverse modalities is still very challenging in terms of *cost* and *flexibility*. For different types of inputs, MLLMs

---

[*]Equal contribution.
[1]Project Page: https://CREMA-VideoLLM.github.io/.

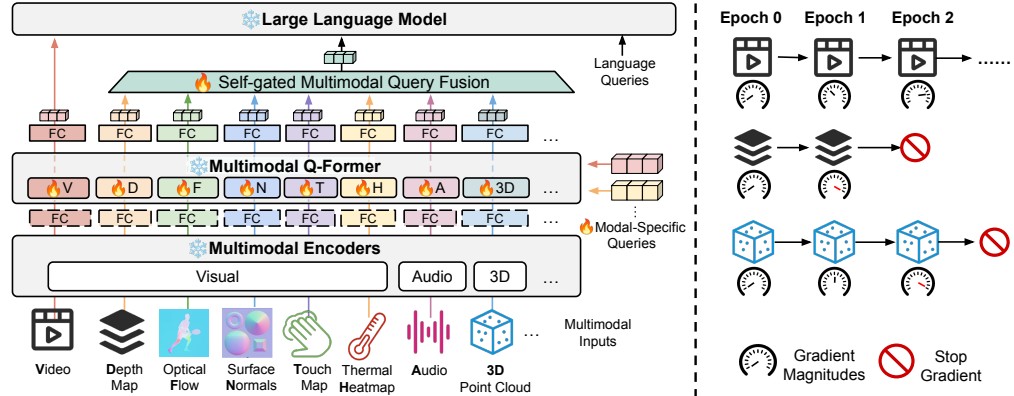

Figure 1: **Overview of the CREMA architecture & training. Left:** Multimodal encoders, Q-former, and LLM are kept frozen in the process. For each modality input, we extract tokens using a corresponding modality-specific adaptation module. Then, we employ the fusion module to blend and compress the obtained multimodal tokens. In the end, the LLM uses modality-fusion tokens to generate responses. **Right:** We present a modality-sequential training and modality-adaptive early exist strategy, further boosting the training efficiency while allowing faster modality adaptation.

have required extremely large computational budgets to update the LLM with individual encoders for modalities. Alternatively, recent efficient MLLMs using separate projection modules (Zhang et al., 2023; Sun et al., 2023a; Li et al., 2023b; Han et al., 2023) provide a more efficient and flexible way for multimodal reasoning. However, as each modality module contains hundreds of millions of parameters for training, this approach is still computationally intensive, and balancing as well as fusing various types of inputs becomes even more complex and costly when more modalities are introduced. Such challenges also exist in very recent pioneering works (Liu et al., 2023c; Panagopoulou et al., 2023; Lu et al., 2023); these models aim to integrate more diverse sensory data for compositional understanding via partial updates to the models, yet still require notable training resources to adapt to different modalities (7B for Unified-IO 2 (Lu et al., 2023)). Moreover, they focus primarily on fixed modality pairs (like 3D-text and visual-text), limiting their adaptability to new data forms and broader applications. This raises the crucial research question: *Can we enable MLLMs to efficiently leverage multiple modalities for Video-Language Reasoning at lower costs, similar to how humans do?*.

In this paper, we present CREMA: a generalizable and highly efficient MLLM framework designed for video-language reasoning with many modalities. The proposed framework enables us to incorporate any new set of modalities, including *video*, *depth map*, *optical flow*, *surface normals*, *audio*, *thermal heatmap*, *touch map*, *3D point cloud*, notably with very few trainable parameters (4∼10M for each new modality) as compared to BLIP-2 (Li et al., 2023b) (∼108M) and SeViLA (Yu et al., 2023a) (∼216M) but higher performance due to assistance of diverse modalities. As illustrated in Figure 1 Left, given a frozen pre-trained vision-language backbone, our approach introduces modality-adaptive modules on top of the Q-Former (Li et al., 2023b) architecture, including linear projectors, low-rank adapters (Hu et al., 2022), and learnable queries. Our parameter-efficient modular design ensures that the pre-trained backbone remains unchanged and enables updates with new modalities and more advanced LLMs in the future without complex architecture changes. To enrich the input modalities, we utilize pre-trained models to extract features from raw videos, e.g., depth maps and optical flow.

Furthermore, despite the effectiveness of our versatile video reasoning framework for multimodal data, handling multiple modalities is not always beneficial; some modalities may be redundant or irrelevant to the reasoning tasks, and optimizing with all modalities simultaneously can lead to a certain deficiency, such as suboptimal convergence of the modality inputs. Besides, the LLM receives longer input contexts, which include token embeddings from all modality queries, resulting in increased computations to produce responses. To address these remaining concerns, we propose a new modality fusion technique that effectively integrates various modality tokens through a novel self-gated attention module and performs alternative modality training within a minibatch, preventing interference in the learning of modality-specific representation. Furthermore, we present a modality-adaptive early exit strategy to bypass the training of a specific modality if it is considered as converged.

This further increases the efficiency of CREMA while allowing faster adaptation and maintaining comparable performance. (Please see Figure 1 Right). In the end, we allow the model to maintain GFLOPs while achieving enhanced reasoning ability, even when the LLM processes many modalities.

We validate CREMA on various video-language reasoning benchmarks assisted by diverse modalities, including VideoQA (NExT-QA (Xiao et al., 2021), PerceptionTest (Pătrăucean et al., 2023)), 3D-VideoQA (SQA3D (Ma et al., 2023)), Audio-VideoQA (MUSIC-AVQA (Li et al., 2022), VG-GSound (Chen et al., 2020)), Touch-VideoQA (Touch&Go (Yang et al., 2022)), Thermal-VideoQA (Thermal-IM (Tang et al., 2023b)). CREMA surpasses other strong multimodal video reasoning baselines, improving fine-tuning performance by **+2.4%** on MUSIC-AVQA, **+2.6%** on SQA3D, and **+0.6%** on NExT-QA while reducing by **90% of the trainable parameters**. CREMA also outperforms general-purpose baselines in the zero-shot setting. We further provide comprehensive analyses of applying CREMA with different LLM, varying sets of modalities, different modality fusion strategies, benefits of adding more modality, and qualitative analysis with input/response visualizations to highlight the efficiency/effectiveness of CREMA in various video-language reasoning across diverse modalities. Our contributions are summarized as follows:

- We propose a **highly efficient and generalizable modality-extensible learning framework**, coined CREMA, which learns multiple modality-adaptive modules to understand given data through augmented senses.
- We present a **novel modality fusion and training design** that efficiently weighs modalities, integrating useful modality features into response generation.
- CREMA's design allows easy embracing of **any new modalities** by adding additional modality-adaptive modules without any need to modify the existing framework.
- We show the efficacy of CREMA on seven video reasoning datasets by achieving better/equivalent performance while **reducing over 90% of trainable parameters** than strong baseline models.
- With CREMA framework, we conduct extensive studies to provide a **comprehensive analysis** of how multimodal information helps video-centric language reasoning tasks.

## 2 RELATED WORKS

**Learning with Multiple Modalities.**    Beyond conventional unimodal learning, leveraging additional modalities, such as visual and audio, in learning models is increasingly popular and has demonstrated remarkable success in solving diverse tasks (Zhu et al., 2024; Liu et al., 2023b; Lu et al., 2023; Moon et al., 2023; Wang et al., 2024a). Vision Language Models (Huang et al., 2023; Li et al., 2023a; Gong et al., 2023; Chen et al., 2023b) are the most prevalent branch of *multimodal learning* that combine vision and language by training on massive data to understand and generate outputs involving visual and text-based information. Audio-Language Models (Chuang et al., 2020; Castellon et al., 2021; Wang et al., 2023) have been proposed for various audio-associated language tasks, *e.g.*, spoken question answering and speech synthesis. Also, 2D-3D Joint Vision (and Language) Models (Li et al., 2020; Hou et al., 2021; 2023; Lei et al., 2024) aim to combine features of both two-dimensional (2D) and three-dimensional (3D) data to interpret and analyze both modalities, allowing for a more comprehensive understanding of visual information. However, these approaches are not scalable to other tasks involving different modality inputs since they focus on handling fine-grained problems with pre-defined modalities.

**Multimodal Large Language Model.**    Very recently, several works propose integrated pipelines using more than two different data sources for general-purpose reasoning (Zellers et al., 2022; Han et al., 2023; Li et al., 2023b; Girdhar et al., 2023; Liu et al., 2023c). MERLOT-REVERSE (Zellers et al., 2022) introduces a new training objective that learns from audio, subtitles, and video frames. Given modality-specific features extracted from the encoder/tokenizer for these inputs, the joint transformer learns to predict the masked text and audio. Therefore, incorporating new types of inputs can be challenging since it requires new pre-training steps for all other modalities. X-InstructBLIP (X-BLIP) (Panagopoulou et al., 2023) integrates various modalities into a framework using frozen LLMs, employing modality-specific Q-Formers as adapters to connect different encoders. However, this method needs to train the individual Q-Former for each modality to enable modality-aligned instruction tuning, which is still resource-intensive. MultiPLY (Hong et al., 2024) is a multisensory embodied LLM for interaction within 3D environments with fixed modality set. X-VILA (Ye et al.,

2024) is an omni-modality model aimed at cross-modality alignment, understanding, and generation. It concentrates on large-scale cross-modality alignment. OneLLM (Han et al., 2023) presents a universal encoder and projection module to align various modalities with language, its flexibility is limited in adapting new modalities, as the pre-trained projection may be impaired with unseen input format. Our CREMA method adopts an efficient and modular approach, using parameter-efficient adapters for each modality, and enhances flexibility in combining any new modalities.

## 3 CREMA: GENERALIZABLE AND EFFICIENT VIDEO-LANGUAGE REASONING VIA MULTIMODAL MODULAR FUSION

We first provide a preliminary of the Q-Former framework for connecting multimodal inputs with the LLM in Section 3.1. Next, we define the problem for compositional VideoQA and introduce our CREMA method for efficient multimodal compositional video reasoning in Section 3.2. Finally, we describe the training and inference process in Section 3.3.

### 3.1 PRELIMINARIES: Q-FORMER

To connect various types of sensory inputs with the LLM, we adopt the Q-Former architecture originally proposed in BLIP-2 (Li et al., 2023b), a transformer (Vaswani et al., 2017) module that bridges the modality encoder and the LLM, similar to Perceiver (Jaegle et al., 2021). It receives modality features $\mathbb{Z}$ from the encoder along with learnable queries $v$ and produces fixed-length tokens $q$ as output. This design enables the Q-Former to compress image tokens into a fixed-length set of query tokens, facilitating efficient processing of video inputs while preserving critical holistic information. It further projects obtained tokens $q$ into the LLM's embedding space via a fully connected layer to make them compatible. In the end, $q$ serves as soft visual prompts (Jia et al., 2022) for the LLM. CREMA method adopts several lightweight form-adaptive modules on top of the Q-Former to integrate knowledge from different data types (*e.g.*, video frames, audio, 3D point cloud, touch map etc.) efficiently.

### 3.2 MULTIMODAL COMPOSITIONAL VIDEO REASONING VIA EFFICIENT MODULAR ADAPTATION AND FUSION

**Multimodal Encoders.** Our proposed method, CREMA, illustrated in Figure 1, aims to generate responses using both language (i.e., questions) and various multimodal inputs (*e.g.*, images, audio, depth data), denoted as $\mathbb{M} = [M_1, M_2, ..., M_n]$, where $n$ represents the number of accessible modalities. Throughout this paper, our CREMA method handles six modalities: video RGB frames, audio, 3D point cloud, optical flow, surface normals, and depth map, in total (up to four different modalities at once), but we note that our approach is able to process a larger number of different data types if needed. We first encode input data for each modality using the corresponding encoder. Here, we adopt several classes of publicly available pre-trained encoders for modalities, which are kept frozen, to improve training efficiency and learning generalization. *Universal encoder* (Girdhar et al., 2023; Han et al., 2023) can be employed as well, but it lacks the flexibility of other new modalities that are not pre-trained before. Next, we add a fully connected layer (*dashed box* in Figure 1) for each data type when dimension misalignment happens. It maps different modality representations into a unified feature space while avoiding incompatibilities between varying encoder architectures. We then obtain a set of multimodal features and in the next step, the Q-Former will extract informative features in $\mathbb{Z}$ into the query embeddings. More details are discussed in Section 4.1 and Section A.2.

**Multimodal Q-Former.** Previous studies (Panagopoulou et al., 2023; Yu et al., 2023b; Zhang et al., 2023; Hong et al., 2023) have shown the capability of the Q-Former architecture to integrate various modalities with LLMs. However, they necessitate the individual Q-former for each modality, leading to significant parameter demands. While Q-Formers are moderately scaled at ∼188 million parameters, which is less than LLMs, this size becomes substantial with increasing modalities. For example, processing six different modalities would require about 1B parameters, highlighting the cost and complexity of scaling the modality with additional Q-Formers.

Hence, to deploy a lightweight, universal module capable of integrating various sensory representations, we introduce *Multimodal Q-Former*. This architecture integrates a Modality-specific

Multi-Query Adapter (MMQA) for each modality. As illustrated in the middle of Figure 1, MMQA consists of Low-Rank Adaptation (LoRA) (Hu et al., 2022) modules[2], learnable queries, and linear projections. The intuitive design of our approach enables efficient and flexible adaptation to any specific modalities. Let $\mathbb{Z}_m$ be the features extracted from the data $M_m$ of the $m^{th}$ modality. The multimodal Q-Former propagates $\mathbb{Z}_m$ with the corresponding MMQA module to capture the relevant information, producing query embeddings $\boldsymbol{q}_m$. Given the learnable input queries $\boldsymbol{v}_m = \boldsymbol{v}_m^0$, we compute a linear projection at layer $i$ containing the modality-specific LoRA as follows:

$$\boldsymbol{v}_m^{i+1} = \boldsymbol{W}\boldsymbol{v}_m^i + \Delta\boldsymbol{W}_m, \tag{1}$$

$$\Delta\boldsymbol{W}_m = \boldsymbol{B}_m\boldsymbol{A}_m \quad \boldsymbol{B} \in \mathbb{R}^{d\times r}, \boldsymbol{A} \in \mathbb{R}^{r\times d}, \tag{2}$$

where $\boldsymbol{W}$[3] represents the original linear projection parameters of the Q-Former. $\Delta\boldsymbol{W}$ indicates a low-rank adapter for $\boldsymbol{W}$ with rank $r \ll d$. Here, $d$ is the feature size of the Q-Former. With the hidden dimension $r = 64$, updating only a small number of parameters of $\Delta\boldsymbol{W}$ for each modality while freezing the Q-Former backbone is sufficient for the model to capture rich modality-specific representation. In addition, the proposed approach can effortlessly integrate new modalities. Upon the arrival of a new modality, our method can simply append appropriate MMQA modules without modifying the existing architecture, ensuring sustained support for previously integrated modalities.

**Self-gated Multimodal Query Fusion.** Our approach, which concatenates modality-adaptive queries from lightweight MMQA modules, efficiently manages multimodal reasoning tasks. However, the LLM faces increased training/inference time and computational costs due to extra input tokens proportional to the number of modalities. To prevent the query token size from growing linearly with each new modality, we introduce a novel self-gated multimodal query fusion module (Assistance Modality Fusion in Figure 1). We define the token embedding of video queries $\boldsymbol{q}_V = \boldsymbol{q}_1$ as a major and others $\boldsymbol{q}_{\backslash V} = \{\boldsymbol{q}_i\}_{i=2}^n$ to be supportive ones. Next, we merge supportive query embeddings through a linear projection layer $\pi(\cdot; \boldsymbol{\theta})$ to match the dimension with $\boldsymbol{q}_V$. Motivated by Ramachandran et al. (2017), we then perform attention on the merged query embeddings $\bar{\boldsymbol{q}}_{\backslash V}$ via self-gated operation and fuse them with $\boldsymbol{q}_V$:

$$\bar{\boldsymbol{q}}_{\backslash V} = \pi\left([\boldsymbol{q}_2; \cdots ; \boldsymbol{q}_n]; \boldsymbol{\theta}\right),$$
$$\widehat{\boldsymbol{q}} = \text{concat}\left(\boldsymbol{q}_V, \left(\text{sigmoid}\left(\bar{\boldsymbol{q}}_{\backslash V}\right) \cdot \bar{\boldsymbol{q}}_{\backslash V}\right)\right), \tag{3}$$

where $[\cdot; \cdots ; \cdot]$ denotes a channel-wise concatenation. This design mirrors human perception in video reasoning tasks, where visual cues are primary but are assisted by other modalities for a richer understanding. In the end, CREMA performs cross-modal reasoning; the model aggregates multimodal query embeddings with the language query $\boldsymbol{l}$ via simple concatenation and feeds the concatenated embeddings into the Large Language Model (LLM) to obtain the final response $a$, such that $a = LLM(\text{concat}(\widehat{\boldsymbol{q}}, \boldsymbol{l}))$.[4] The proposed design of modality fusion in CREMA successfully reduces computational costs when incorporating new modalities, achieving comparable or even enhanced performance compared to CREMA without the modality fusion.

### 3.3 MODALITY-SEQUENTIAL AND MODULAR TRAINING OF CREMA

Leveraging the proposed highly efficient modality-specific multi-query adapter (MMQA) and the self-gated multimodal query fusion approach, CREMA effectively processes a wide range of sensory input types without requiring tailored architecture designs. However, different input modalities possess distinct characteristics and varying quantities of information. Directly optimizing the model on diverse modality inputs may lead to suboptimal convergences, suffering from over/under-fitting on dominant/insignificant data types. To address the susceptibility of optimization when training on various modality inputs simultaneously, as shown in Figure 1 Right, we propose a simple yet effective remedy: ***modality-sequential modular training*** and ***adaptive early exit***. Inspired by the alternating update of multimodal representation learning (Zhang et al., 2024), we propose a sequential, modality-adaptive optimization process for each iteration. Instead of performing a joint back-propagation step for all modalities simultaneously, we decompose it into sequential modality-specific updates.

---

[2]We implement LoRA modules at the *query* and *value* linear projections for each self-attention layer.

[3]For the rest of the paper, unless otherwise stated, we omit the layer index for readability.

[4]To let the LLM be aware of the difference of each modality, we insert modal-specific prefix tokens before modality queries. We omit the notation from the equation for simplicity.

Table 1: **Fine-tuning Results on Audio-Video Question Answering (MUSIC-AVQA).** We report simple notations for each modality and question type: **V**: *Video RGB frames*, **A**: *Audio*, **F**: *optical Flow*, **D**: *Depth*, and **N**: *surface Normalization*. **Cnt.**: *Counting*, **Com.**: *Comparative*, **Loc.**: *Location*, **Ext.**: *Existential*, and **Tem.**: *Temporal*. We **bold** the best and underline the second-best numbers.

| Method | Modality | Audio | | | Visual | | | Audio-Visual | | | | | | Avg. | Trainable Params. | GFLOPs |
|---|---|---|---|---|---|---|---|---|---|---|---|---|---|---|---|---|
| | | Cnt. | Com. | Avg. | Cnt. | Loc. | Avg. | Ext. | Loc. | Cnt. | Com. | Tem. | Avg. | | | |
| AVQA (Li et al., 2022) | V, A | 80.3 | 60.0 | 77.3 | 74.5 | 77.8 | 76.1 | 81.4 | 68.7 | 69.9 | 64.6 | 67.1 | 70.9 | 73.5 | 18M | - |
| LAVISH (Lin et al., 2023) | V, A | 85.6 | **65.9** | 81.4 | 80.2 | 81.1 | 80.6 | 84.6 | 69.2 | 78.8 | 65.6 | 69.1 | 73.8 | 76.9 | 21M | - |
| BLIP-2 (Li et al., 2023b) | V | 86.7 | 58.5 | 76.3 | 87.2 | **93.7** | **90.5** | 81.5 | 72.0 | 81.3 | 64.3 | 70.1 | 74.2 | 78.9 | 108M | 1.30K |
| | V, A | 86.3 | 58.4 | 76.0 | **87.6** | 93.0 | 90.3 | 80.4 | 68.3 | 82.6 | 63.8 | 69.8 | 73.5 | 78.4 | 216M | 1.78K |
| | V, A, F | 86.0 | 59.9 | 76.4 | 84.9 | 92.5 | 88.8 | 81.8 | 71.4 | 79.7 | 65.1 | 68.4 | 73.6 | 78.1 | 324M | 3.21K |
| ☕CREMA (Ours) | V | 88.3 | 60.6 | 82.3 | 84.4 | 85.2 | 84.8 | 84.8 | 71.7 | 80.8 | 63.8 | 70.6 | 74.6 | 78.7 | 4M | 1.30K |
| | V, A | **89.0** | 61.4 | 83.0 | 84.7 | 85.0 | 84.8 | 84.4 | 73.2 | 84.8 | 63.2 | 71.3 | 75.6 | 79.4 | 9M | 1.78K |
| | V, A, F | 87.1 | 61.0 | 81.5 | 84.2 | 90.3 | 87.2 | 83.4 | 74.2 | 82.6 | **68.7** | 71.7 | 76.4 | 80.5 | 21M | 1.81K |
| | V, A, F, D, N | 88.3 | 64.7 | **83.2** | 86.2 | 91.3 | 88.7 | **84.9** | **74.9** | **85.2** | 68.6 | **73.5** | **77.7** | **81.7** | 38M | 1.84K |

This approach selectively updates the trainable weights corresponding to the target modality input, ensuring more focused and efficient learning for each modality. Specifically, given the minibatch data $M = (M_1, ..., M_n)$ containing $n$ modalities, we propagate $M$ through the model but update only the trainable modules corresponding to the target modality $M_m$ (*i.e.*, the MMQA module for $m$ and the fusion module). Note that this sequential training within the minibatch helps the model remain robust to the order of modalities during training. Furthermore, we extend this to a modality-adaptive early exit strategy, allowing the model to bypass the training of a specific modality if it is considered as converged. We use the average gradient magnitude of the MMQA module weights as a metric for the early exit of each modality input. Specifically, at each training epoch $j + 1$, we determine whether the model exits the training of the target modality if it satisfies the following equation: $\bar{g}[j + 1] > \tau \cdot \text{average}(\bar{g}[: j])$[5], where the temperature parameter $\tau$ and $\bar{g} = [\bar{g}_1, ..., \bar{g}_{j+1}]$ denotes the list of the averaged gradient magnitudes obtained at the end of each training epoch. In the end, CREMA effectively learns rich information from various modalities for video-language reasoning, while mitigating unnecessary over-convergence or imbalanced training across different modalities.

## 4 EXPERIMENTS

In this section, we first outline the overall experimental setup in Section 4.1 and show the results of the proposed CREMA on various cross-modal Video QA & reasoning tasks in Section 4.2. We further provide more insights about the MLLM backbone, training strategy, design of the fusion module, and the impact of new modalities across tasks in Section 4.3. More experiments on extra datasets (VGGSound (Chen et al., 2020) and PerceptionTest (Pătrăucean et al., 2023)), training memory comparison, the impact of sequential training, LoRA rank, MMQA initialization, and qualitative examples are included in Appendix (Section B).

### 4.1 EXPERIMENTAL SETUP

**Datasets & Benchmarks.** We evaluate CREMA on the following video reasoning and QA tasks: *SQA3D* (Ma et al., 2023), *MUSIC-AVQA* (Li et al., 2022), and *NExT-QA* (Xiao et al., 2021). We further evaluate CREMA on TouchQA and ThermalQA collected by ourselves based on public video-touch (Touch&Go (Yang et al., 2022)) and video-thermal data (Thermal-IM (Tang et al., 2023b)). See Appendix (Sections A.1 and A.3) for more details.

**Implementation Details.** **Pre-trained Visual Experts**: We employ frozen pre-trained visual experts to extract modalities features from raw videos. Specifically, we use ZoeDepth (Bhat et al., 2023), Unimatch (Xu et al., 2023), and NLL-AngMF (Bae et al., 2021) to estimate depth, flow, and normals, respectively. **Modality Encoder:** We use frozen modality-specific encoders to encode each modality to embedding space. We adopt ViT-G (Sun et al., 2023b) for visual (frames, depth, norm, flow, touch, and thermal), BEATs (Chen et al., 2023a) for audio, and follow data extraction in 3D-LLM (Hong et al., 2023) and ConceptFusion (Jatavallabhula et al., 2023) for 3D point cloud. See Appendix (Section A.2) for details. **Baselines & Model Implementation**: We extend 3D-LLM and

---

[5]We omit the target modality index for brevity.

Table 2: **Fine-tuning Results on 3D Situated Question Answering (SQA3D).** **V**: *Video RGB frames*, **V***: *Bird-Eye View image*, **P**: *3D Point cloud*, **D**: *Depth*, and **N**: *surface Normalization*.

| Method | Modality | What | Is | How | Can | Which | Others | Avg. | Trainable Params. | GFLOPs |
|---|---|---|---|---|---|---|---|---|---|---|
| MCAN (Yu et al., 2019) | V* | 28.8 | 59.6 | 44.0 | 68.3 | 40.7 | 40.4 | 43.4 | 56M | - |
| ClipBERT (Lei et al., 2021) | V | 30.2 | 60.1 | 38.7 | 63.3 | 42.4 | 42.7 | 43.3 | 135M | - |
| ScanQA (Azuma et al., 2022) | P | 31.6 | 63.8 | 46.2 | 69.5 | 43.8 | 45.3 | 46.5 | 38M | - |
| 3D-LLM (Hong et al., 2023) | V | 45.1 | 62.7 | 48.6 | 63.3 | 45.8 | 49.8 | 51.4 | 108M | 1.30K |
|  | V, P | 47.7 | 61.0 | 49.0 | 63.6 | 49.0 | 49.6 | 52.3 | 216M | 1.67K |
|  | V, P, D, N | 45.2 | 62.4 | 44.8 | 64.2 | 45.8 | 49.4 | 52.0 | 434M | 5.47K |
| ☕ CREMA (Ours) | V | 44.9 | 62.1 | 48.1 | 67.7 | 48.4 | 49.8 | 51.8 | 4M | 1.30K |
|  | V, P | 46.2 | 63.6 | 46.4 | 63.0 | 48.7 | 50.1 | 52.1 | 8M | 1.69K |
|  | V, P, D, N | 47.6 | 66.0 | 51.0 | 65.1 | 48.4 | 56.4 | 54.6 | 38M | 1.83K |

Table 3: **Fine-tuning Results on Video Question Answering (NExT-QA).** Question types are abbreviated as: **P.&N.**: *Prev & Next*, **Pre.**: *Present*, **Cnt.**: *Count*, **Loc.**: *Location*, and **Otr.**: *Other*.

| Methods | Modality | Causal | | | Temporal | | | Descriptive | | | | Avg. | Trainable Params. | GFLOPs |
|---|---|---|---|---|---|---|---|---|---|---|---|---|---|---|
|  |  | How | Why | Avg. | P.&N. | Pre. | Avg. | Cnt. | Loc. | Otr. | Avg. |  |  |  |
| LLaMA-VQA (7B) (Ko et al., 2023) | V | - | - | 72.7 | - | - | 69.2 | - | - | - | 75.8 | 72.0 | 5M | - |
| Mirasol3B (Piergiovanni et al., 2023) | V | - | - | - | - | - | - | - | - | - | - | 73.2 | 3B | - |
| SeViLA (Yu et al., 2023a) | V | 71.3 | 75.3 | 74.2 | 67.8 | 71.7 | 69.4 | 67.2 | 91.8 | 85.2 | 81.3 | 73.8 | 216M | - |
| BLIP-2 (Li et al., 2023b) | V | 69.9 | 73.9 | 72.9 | 65.4 | 71.9 | 68.1 | 64.9 | 91.8 | 80.3 | 81.2 | 72.6 | 108M | 1.30K |
|  | V, F | 68.8 | 74.0 | 72.6 | 65.8 | 71.1 | 68.0 | 64.9 | 92.8 | 81.3 | 81.9 | 72.6 | 216M | 2.21K |
|  | V, F, D | 70.8 | 74.2 | 73.3 | 65.0 | 71.4 | 67.6 | 61.5 | 93.2 | 81.6 | 81.4 | 72.7 | 324M | 5.03K |
|  | V, F, D, N | 71.7 | 74.2 | 73.5 | 65.7 | 72.6 | 68.5 | 65.5 | 92.5 | 81.9 | 82.1 | 73.3 | 432M | 6.12K |
| ☕ CREMA (Ours) | V | 67.3 | 73.9 | 72.1 | 63.0 | 70.2 | 65.9 | 64.9 | 93.2 | 80.3 | 81.6 | 71.6 | 4M | 1.30K |
|  | V, F | 69.3 | 74.1 | 72.8 | 64.4 | 70.5 | 66.9 | 67.2 | 92.8 | 80.9 | 82.2 | 72.4 | 8M | 2.22K |
|  | V, F, D | 70.4 | 74.4 | 73.3 | 66.6 | 72.4 | 69.0 | 61.0 | 92.5 | 81.0 | 80.8 | 73.2 | 20M | 2.34K |
|  | V, F, D, N | 71.3 | 75.5 | 74.4 | 67.3 | 72.5 | 69.4 | 66.1 | 92.9 | 79.7 | 81.6 | 73.9 | 28M | 2.47K |

BLIP-2 with **individual Q-Formers** for each new modality as our baseline. We fully fine-tune these Q-Formers. Our Multimodal Q-Former is initialized from BLIP-2 pre-trained one. We set 64 LoRA rank and 32 query tokens for all MMQA modules. More details in Appendix (Sections A.4 and A.5).

## 4.2 MAIN EXPERIMENTAL RESULTS

**MUSIC-AVQA:** CREMA achieves superior audio-video reasoning ability. In Table 1, recent parameter-efficient approaches, AVQA and LAVISH, perform reasonably well on audio and video QA tasks (MUSIC-AVQA), but are less impactful due to their restricted language capability. BLIP-2 achieves higher accuracy by training modality-specific Q-Formers with a powerful language model, FLAN-T5XL. However, it fails to incorporate multiple modalities, and degrades audio-video reasoning ability when combining *V* modality with other modalities: *A*, *F*, *D*, and *N*. Our method constantly improves the average accuracy with more modality, outperforming LAVISH (**+4.8%***p*) and BLIP-2 (**+2.4∼3.6%***p*), by using only **6.4∼11.7%** trainable parameters compared to BLIP-2.

**SQA3D:** Our method is significantly efficient yet outperforms publicly available, strong MLLM baselines on 3D-associated video reasoning. As shown in Table 2, we evaluate the fine-tuning performance on SQA3D, and CREMA with video frame inputs (*V*) obtains improved accuracy compared to baselines on single modality inputs, MCAN, ClipBERT, and ScanQA. We also measure the performance of 3D-LLM, a strong multimodal learning method that shares almost all of its structures with BLIP-2, except for the 3D encoder. Although 3D-LLM enhances its performance by integrating multiple modalities, this brings a considerable increase in parameters to update multiple Q-Formers for each modality. Meanwhile, our CREMA method with *V, P, D, N* modalities surpasses all baselines, achieving the best average accuracy by updating only proposed MMQA modules, which uses ∼**91.2%** fewer parameters (38M) for training than 3D-LLM (434M).

**NExT-QA:** CREMA consistently achieves superior performance against strong vision-language reasoning methods on the NExT-QA dataset. As shown in Table 3, SeViLA, BLIP-2, and CREMA adopt Flan-T5XL (3B) but achieve reasoning capabilities comparable to the LLAMA-7B model. CREMA with *V* obtains a slightly lower fine-tuning performance compared to BLIP-2 since they perform fine-tuning of the entire Q-Former framework. However, the ability of our proposed

Table 4: Fine-tuning Results on **TouchQA** and **ThermalQA**. T: Touch map, H: thermal Heatmap.

| | Method | Modality | Acc. | Tr. Params. |
|---|---|---|---|---|
| **TouchQA** | CMC | V, T | 44.3 | 12M |
| | BLIP-2 | V | 78.2 | 108M |
| | | V, T | 77.4 | 216M |
| | ☕CREMA (Ours) | V | 78.0 | 4M |
| | | V, T | 79.1 | 8M |
| | | V, T, N | **79.3** | 20M |
| **ThermalQA** | CMC | V, H | 40.3 | 12M |
| | BLIP-2 | V | 55.2 | 108M |
| | | V, H | 54.4 | 216M |
| | ☕CREMA (Ours) | V | 54.9 | 4M |
| | | V, H | 56.2 | 8M |
| | | V, H, D | **56.7** | 20M |

Table 5: **Zero-shot Evaluation** on Multimodal Compositional QA tasks (SQA3D and MUSIC-AVQA).

| Method | Modality | Acc. | Total Params. |
|---|---|---|---|
| **SQA3D** | | | |
| Unified QA | P | **41.0** | 11.0B |
| GPT-3 | P | **41.0** | 175.0B |
| 3D-LLM | P | 36.9 | 3.1B |
| OneLLM | P | 34.5 | 7.8B |
| | V | 39.4 | 7.8B |
| | V, P | 37.9 | 7.8B |
| ☕CREMA (Ours) | P | 37.3 | 3.1B |
| | V | 39.6 | 4.1B |
| | V, P | 40.0 | 4.1B |

| Method | Modality | Acc. | Total Params. |
|---|---|---|---|
| **MUSIC-AVQA** | | | |
| X-InstructBLIP | A | 22.7 | 13.2B |
| | V | 43.5 | 14.1B |
| | V, A | 44.5 | 14.4B |
| OneLLM | A | 34.8 | 7.8B |
| | V | 48.4 | 7.8B |
| | V, A | 42.3 | 7.8B |
| ☕CREMA (Ours) | A | 31.0 | 3.2B |
| | V | 51.0 | 4.1B |
| | V, A | **52.6** | 4.2B |

framework to incorporate a variety of new modalities enhances its compositional understanding: CREMA with *V, F, D, N* surpasses BLIP-2 and SeViLA with **87~94% less parameters** for training.

**TouchQA & ThermalQA:** To further demonstrate the generalizability of our framework over unique/rare sensory inputs, we evaluate CREMA on *video-touch* and *video-thermal* QA tasks. In Table 4, BLIP-2 shows competitive performance but struggles to integrate data from different modalities for reasoning, resulting in degraded performance when using two modalities. In contrast, CREMA achieves superior performance using only 20M trainable parameters. This advantage is particularly significant when compared to the parameter-efficient multimodal reasoning method, CMC, showing a considerable margin of performance improvement despite similar trainable parameters.

**Zero-shot Evaluation:** In addition to the fine-tuning evaluation, CREMA method also achieves superior zero-shot performance on compositional video reasoning. We perform zero-shot evaluations on SQA3D and MUSIC-AVQA in Table 5. Note that Unified QA (Khashabi et al., 2020) and GPT-3[6] with caption generated from 3D point cloud input perform well, attributed to their considerable model size and pre-trained data. We also observe that OneLLM (Han et al., 2023), a universal multimodal reasoning framework equipped with Llama2-7B (Touvron et al., 2023) for the LLM backbone, degenerates the performance when integrating different modalities: *V, P*. On the other hand, CREMA demonstrates a distinct advantage in zero-shot compositional reasoning across modalities, improving performance when combining video frames and 3D point clouds. We also compare with X-InstructBLIP (Panagopoulou et al., 2023), a strong reasoning framework integrating various modalities with modality-specific Q-Formers as adapters to connect different encoders, and OneLLM on audio-video reasoning tasks. As shown, our method outperforms both X-BLIP (13B) and OneLLM (7.8B), by **+8.1%**$p$ and **+10.3%**$p$ (*V, A*).

**CREMA with Different MLLM Backbone:** We further built a new version of CREMA with VideoChat2 (Li et al., 2024) using stronger Mistral-7B as an LLM backbone and processed more frames (16 frames following the VideoChat2 setting). As shown in Table 6, we observed that our CREMA achieved impressive fine-tuning performance (79.4%) outperforming other strong video-LLMs by incorporating optical flow (F) and depth map (D) on NExT-QA while using a remarkably smaller number of trainable parameters, demonstrating the generalization ability across MLLM and efficiency of our proposed framework. We leave the CREMA with even stronger MLLMs (e.g., BLIP-3 (Xue et al., 2024), Qwen-VL (Wang et al., 2024b)) with more diverse modalities for further studies.

Table 6: Comparison between **CREMA with Mistra-7B** and other popular Video-LLMs with comparable size LLMs. ∗: our reproduced.

| Model (Modality) | LLM | Acc. | Tr. Params. |
|---|---|---|---|
| Video-LLaMA (V) | Vicuna-7B | 60.6 | 216M |
| Video-LLaVA (V) | Vicuna-7B | 66.3 | 7B |
| VideoChat2 (V) | Vicuna-7B | 68.6 | >200M |
| LLaMA-VQA (V) | LLaMA2-7B | 72.0 | 5M |
| MotionEpic - VoT (V) | Vicuna-7B | 76.0 | >100M |
| LLaVA-NeXT (V) | Qwen1.5-7B | 78.2 | 7B |
| VideoChat2* (V) | Mistral-7B | 78.4 | >200M |
| CREMA (V, F) | Mistral-7B | **78.9** | 21M |
| CREMA (V, F, D) | Mistral-7B | **79.4** | 45M |

---

[6]We borrow the results of GPT-3 from the official technical report in (Ma et al., 2023).

## 4.3 QUANTITATIVE ANALYSIS

We have extensively validated the versatility of the proposed CREMA approach over a broad range of video-related reasoning tasks, assisted by various supportive modalities. Now, we provide in-depth analyses addressing the following two research questions:

**RQ 1: Why did the other baseline struggle to enhance video reasoning with more modality?**
To enable an LLM to comprehend information from non-linguistic modality inputs, we need to project them into a unified language embedding space. However, this becomes increasingly challenging as the number of different modalities grows, each representing distinct attributes and sensory information. This issue is observed in models like 3D-LLM and BLIP-2 (See their degraded or on-par performance when injecting new modalities in Tables 1 and 2), which introduce individual attention modules to produce modality-specific query embeddings used for LLM reasoning. Due to the lack of a shared representation space and an intelligent fusion design, these embeddings often suboptimally converge, making them less compatible with other modalities in terms of representation distribution and resulting in insufficient compositional video reasoning capabilities. Furthermore, as discussed in Section 3.3, employing a universal optimization policy for learning different modality inputs can cause the model to over-converge on a few dominant modalities, which is also another critical issue in solving the *many-modal* video reasoning problem.

**RQ 2: How does CREMA address challenges and help video reasoning with more modalities?**
We note that CREMA exhibits improved compositional reasoning abilities using more diverse modalities, evident in higher average performance. The model benefits from the following crucial innovations. 1) we compel a regularization effect on the model through parameter-efficient updates, often leading to a stable and better generalization than fine-tuning large models (Zhao et al., 2021; Ding et al., 2022; Fu et al., 2023). In addition, the modular adaptation from the unified backbone using MMQA and the proposed novel modality fusion approach allow the model to produce compatible yet informative modality token embeddings sampled within a stable multimodal representation space. 2) Our proposed modality-sequential training and modality-adaptive early exit approach help CREMA to optimize multiple modality inputs effectively. In particular, we demonstrate the efficacy of our proposed modality-adaptive early exit strategy. Let the early stop indicator be $I_{es} = \bar{g}[j+1]/\tau \cdot \text{average}(\bar{g}[:j])$, implying that CREMA exits to learn the specific modality data once the corresponding $I_{es}$ reaches 1. We observe that our adaptive early exit reduces training time/computations by $25 \sim 60\%$, as demonstrated in Table 7 and Figure 2, without necessitating additional complex computations or calibration data. Furthermore, CREMA with a modality-adaptive early exit facilitates faster model convergence, which is evident in a **performance increase of 1.0 $\sim$ 1.1%** in both datasets, compared to CREMA trained for a similar number of epochs. Please see Section B.4 for the ablation of the modality-sequential training.

**RQ3: Ablations for the modality fusion module of CREMA.** As the architectural design of the fusion module affects the fusion quality of token embeddings, we investigate our CREMA with different fusion strategies: given a concatenated multimodal token embedding $\mathbb{Q}$ obtained by a multimodal Q-Former (i.e., *concat*), *Linear* reduces the token size of $\mathbb{Q}$ through a linear projection, *Mixture-of-Experts (MoE)* adopts a MoE layer inside Q-Former to extract a few token embeddings, and *Cross-Attention* adopts extra prompts as an input and computes the cross-attention with $\mathbb{Q}$. As shown in Table 8, our proposed *self-gated* fusion module achieves competitive performance with *Concat*, while other variants decrease the average accuracy despite incorporating additional modalities besides *video*. Also, it performs efficiently compared to *Cross-attention* and *Linear*, requiring more computationally expensive operations to combine modality. We expect that the modality fusion via *MoE* or *Cross-attention* alters the original model architecture of the Q-former through additional parametric layers. Thus, it struggles to unlock the capabilities of those powerful designs under the parameter-efficient fine-tuning setup and the limited scale of downstream data.

**RQ4: The impact of new modalities on easy/hard questions.** We further delve into how adding extra modalities beyond video RGB frames (V) can enhance video reasoning problems. Following the previous work (Buch et al., 2022), which splits the dataset into the easy/hard groups based on the performance of the reference model to find subsets requiring less/more modality information, we classify question inputs in compositional video reasoning tasks (SQA3D and MUSIC-AVQA) based on the zero-shot performance of CREMA method with only V; i.e., if the model predicts correctly $\rightarrow$ *easy*, otherwise $\rightarrow$ *hard*, indicating that input examples in *hard* may need additional knowledge to find appropriate answers. After that, we fine-tune our CREMA and test on obtained

Table 7: **Ablation of Modality-adaptive Early Exit** on MUSIC-AVQA (vs. BLIP-2 w/ V, A, F) and SQA3D (vs. 3D-LLM w/ V, P, D, N).

| Method | Acc. | Tr. Params. | Avg. Epochs | Acc. | Tr. Params. | Avg. Epochs |
|---|---|---|---|---|---|---|
| BLIP-2 (or) | 77.8 | 324M | 8 | 51.2 | 432M | 10 |
| 3D-LLM | 78.1 | 324M | 20 | 52.0 | 432M | 20 |
| ☕ CREMA | 79.3 | 38M | 8 | 52.8 | 38M | 10 |
| (Ours) | **80.5** | 38M | 20 | **54.6** | 38M | 20 |
| +Adaptive | 80.3 | 38M | ∼8 | 53.9 | 38M | ∼10 |
| Early Exit | **80.5** | 38M | ∼13 | 54.3 | 38M | ∼15 |

Table 8: **Average accuracy & GFLOPs (on NExT-QA) of our method with different modality fusion strategies.** We use *V, F, D, N* on NExT-QA and *V, P, D* on SQA3D. *Concat* indicates that we concatenate multimodal query tokens.

| Fusion | NExT-QA | SQA3D | GFLOPs |
|---|---|---|---|
| Video-Only (V) | 71.6 | 51.8 | 1.30K |
| Concat | 73.5 (+1.9) | 53.0 (+1.2) | 6.14K |
| Cross-Attention | 70.5 (−1.1) | 49.8 (−2.0) | 2.33K |
| Linear | 71.1 (−0.5) | 51.3 (−0.5) | 1.87K |
| MoE | 72.0 (+0.4) | 51.1 (−0.7) | 1.02K |
| **Self-Gated (Ours)** | **73.9** (+2.3) | **54.6** (+2.8) | 2.47K |

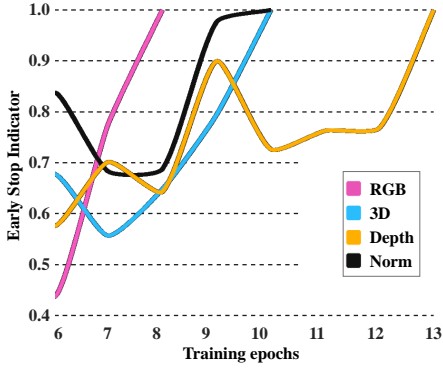

Figure 2: **Modality-adaptive Early Exit** of CREMA on SQA3D. CREMA stops to update MMQA modules for a specific modality once the corresponding indicator value reaches 1.0.

Table 9: **Acuracy of CREMA method on easy and hard questions** across datasets and modalities.

| Modality | Easy Acc. | Hard Acc. |
|---|---|---|
| | **SQA3D** | |
| V | 75.0 | 37.2 |
| V, P | **75.8** (+0.8) | 37.3 (+0.1) |
| V, P, D, N | 72.6 (−2.4) | **42.1** (+4.9) |
| | **MUSIC-AVQA** | |
| V | 85.7 | 71.2 |
| V, A | 87.2 (+1.5) | 71.4 (+0.2) |
| V, A, F, D, N | **87.9** (+2.2) | **75.2** (+4.0) |
| | **NExT-QA** | |
| V | **90.6** | 41.5 |
| V, D | 90.2 (−0.4) | 43.1 (+1.6) |
| V, F, D | 90.5 (−0.1) | 44.7 (+3.2) |
| V, F, D, N | 89.1 (−1.5) | **50.0** (+8.5) |

subsets. As shown in Table 9, adding new modalities brings improvement over both *easy* and *hard* subsets. However, performance gain on the *easy* subset is less effective as it is already dominant to the video frame inputs, whereas information from additional modalities benefits the prediction of the *hard* (**+4.9**%p on SQA3D and **+4.0**%p on MUSIC-AVQA). In NExT-QA, adding new modalities marginally decreases the *easy* subset, but significantly boosts the *hard* (**+8.5**%p). It indicates that leveraging extra modalities can be an effective modality augmentation strategy, mitigating overfitting. Furthermore, CREMA method can be an efficient tool to determine modality importance for video reasoning benchmark designs.

## 5 CONCLUSION

This paper introduces CREMA, an efficient and powerful framework for multimodal compositional video reasoning. We introduce parameter-efficient modality-adaptive modules atop a multimodal Q-former to seamlessly incorporate any new modalities like video, optical flow, audio, 3D point cloud, etc. Since our CREMA method does not require modifying the backbone architectures, we can easily upgrade our framework with new and stronger language models in the future without damaging its ability on existing modalities. We demonstrate the efficacy of our method on various multimodal QA benchmarks, surpassing baselines' performance with a notable reduction in trainable parameters. We present a multimodal fusion module and training strategy to keep low computational costs in LLM while achieving better performance, when we integrate more modalities.

## ETHICS STATEMENT

The intended use of CREMA is to conduct video-language reasoning with the help of diverse assistance modalities. This does not have any particular potential for misuse beyond the general potential for AI technology to be used in harmful ways. Because it is based on MLLM to answer video questions, CREMA has the potential to hallucinate. Note that potential is shared widely in

the MLLMs/Video-LLMs (Li et al., 2023b; 2024). The fact that CREMA answers questions with diverse modality augmentation, enhances the system performance while delimiting model overfitting thus mitigating model hallucination. This is crucial to build reliable, trustworthy video-language reasoning systems that assist educational AI.

## ACKNOWLEDGEMENT

We thank the reviewers, and Jaemin Cho, Abhay Zala, Han Lin, Yi-Lin Sung, and Ziyang Wang for their valuable feedback and input for the paper. This work was supported by the National Institutes of Health (NIH) under other transactions 1OT2OD038045-01, ARO Award W911NF2110220, DARPA KAIROS Grant FA8750-19-2-1004, ONR Grant N00014-23-1-2356, DARPA ECOLE Program No. HR00112390060, DARPA MCS Grant N66001-19-2-4031, and NSF-AI Engage Institute DRL211263. The views, opinions, and/or findings contained in this article are those of the authors and not of the funding agency.

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

APPENDIX

In this Appendix, we present the following:

- Additional information about our experimental setup (Section A), including benchmarks details (Section A.1), details of multimodal encoder and visual expert models (Section A.2), MMQA module pre-training data pre-processing (Section A.3), baseline implementation details (Sec. A.4), and CREMA implementation details (Sec. A.5).

- Additional experiments (Section B), including fine-tuning results on VGGSound&PerceptionTest (Section B.1), comparison of training memory (Section B.2), zero-shot per-task performance on MUSIC-AVQA (Section B.3), the impact of MMQA pre-training (Section B.5), the impact of LoRA rank (Section B.6), more qualitative visualization (Section B.7), and extra discussion on model design (Section B.8).

- Limitation and broader impact of this work (Section C), as well as license information (Section D) for the datasets, codes, and models that we used in this paper.

- License information (Section D) for the datasets, codes, and models that we used in this paper.

## A EXPERIMENTAL SETUP

In this section, we present additional information on the used datasets/benchmarks (Sec. A.1), multimodal encoder and visual expert models (Section A.2), baseline implementation (Sec. A.4), and CREMA implementation details (Sec. A.5).

### A.1 BENCHMARK AND DATASET

We evaluate the CREMA framework on three video reasoning and QA tasks, focusing on both conventional VideoQA (requires video and language) and compositional VideoQA (requires video, language and other modalities). These include: (1) **SQA3D** (Ma et al., 2023): Another compositional Video QA task, requiring the understanding of video, 3D scenes, and text. Designed for 3D situated QA, it includes 33K questions and 650 3D scenes corresponding to ego-centric videos. We apply extra depth maps to it and report results on the test part following (Hong et al., 2023). (2) **MUSIC-AVQA** (Li et al., 2022): A compositional Video QA benchmark that involves reasoning across video, audio, and text. This dataset contains 9288 videos and 45K questions. We follow X-InsturctBLIP (Alayrac et al., 2022) to evaluate our CREMA and on other baselines on the high-quality real video part. We enhance it with optical flow as an extra input and report our findings on the test set. (3) **NExT-QA** (Xiao et al., 2021): A conventional Video QA benchmark for causal and temporal reasoning with video and text inputs. It consists of 5440 videos and 52K questions. We include optical flow, depth map, and surface normals extracted from raw videos as additional modalities. Our results are based on the validation partition following previous work (Yu et al., 2023a). **(4) ThouchQA:** we build a question and answering dataset based on Touch&Go (Yang et al., 2022). As the original data only contains material class annotations, we reformulate the classification task as a QA task, by asking the model: *What material is the person touching?*. Our TouchQA dataset contains 714 training data and 3212 test

Table 10: Baseline models fine-tuning hyperparameters.

| Model (Dataset) | Modality | # Frames | Batch Size per GPU | Learning Rate | Warmup | Epoch | Gradient Accumulation Step |
|---|---|---|---|---|---|---|---|
| 3D-LLM (SQA3D) | V | 4 | 16 | 3e-05 | 1000 | 20 | 2 |
| | V, P | 4 | 16 | 3e-05 | 1000 | 20 | 2 |
| | V, P, D, N | 4 | 8 | 3e-05 | 1000 | 20 | 2 |
| BLIP-2 (MUSIC-AVQA) | A | 4 | 8 | 3e-05 | 1000 | 20 | 2 |
| | V | 4 | 8 | 3e-05 | 1000 | 20 | 2 |
| | V, A | 4 | 8 | 3e-05 | 1000 | 20 | 2 |
| | V, A, F | 4 | 8 | 3e-05 | 1000 | 20 | 2 |
| BLIP-2 (NExT-QA) | V | 4 | 16 | 3e-05 | 1000 | 10 | 1 |
| | V, F | 4 | 8 | 3e-05 | 1000 | 10 | 2 |
| | V, F, D | 4 | 4 | 3e-05 | 1000 | 10 | 4 |
| | V, F, D, N | 4 | 4 | 3e-05 | 1000 | 10 | 4 |

data. **(5) ThouchQA:** Similar to the TouchQA dataset, we build the ThermalQA dataset on public video-thermal heatmap dataset, Thermal-IM (Tang et al., 2023b). Thermal-IM contains action labels in each video and was originally designed to predict human pose 3 seconds ago according to both video and thermal heatmap. We reformulate it as a QA task as well by asking the model: *What action might have occurred before this video?* and the answer is action label. Our Thermal dataset contains 1131 training data and 391 test data. **(6) PerceptionTest (Pǎtrǎucean et al., 2023):** a multimodal benchmark designed to comprehensively evaluate the perception and reasoning skills of multimodal video models. We use the multi-choice QA part of this benchmark, which contains 1955 train data and 5260 validation data. **(7) VGGSound (Chen et al., 2020):** is a large-scale audio-video dataset. It contains over 200K videos with audio sounds. It was originally for audio-video classification. We formulated the classification tasks as the the open-ended QA task over 300 audio classes.

## A.2 MULTIMODAL ENCODERS AND VISUAL EXPERT MODELS

We apply multiple encoders to encode multimodal raw input as discussed in Section 3.2. For visual inputs (video RGB frames, depth map, optical flow, and surface normals), we follow BLIP-2 (Li et al., 2023b) and X-InstructBLIP (Panagopoulou et al., 2023) to utilize ViT-G (Sun et al., 2023b) to encoder visual information. As depth map, optical flow, and surface normals raw data are with different channel numbers to the ViT model required, We transform those extra visual information to the RGB domain first to adapt the ViT model. For audio, we use the same BEATs$_{ITER3+}$ (Chen et al., 2023a) encoder as in X-InstructBLIP (Panagopoulou et al., 2023). For the 3D point cloud, we follow data preprocessing in 3D-LLM (Hong et al., 2023) and ConceptFusion (Jatavallabhula et al., 2023) that first extract pixel-aligned dense features for rendered images features and then fuse 2D features into 3D maps using gradslam (Jatavallabhula et al., 2019).

We employ plug-and-play frozen experts to extract diverse modalities features, including depth map, optical flow, and surface normals, from raw videos. For optical flow estimation, we utilize the SotA Unimatch (Xu et al., 2023) model (GMFlow-scale2-regrefine6-mixdata). For depth map estimation, we leverage the SotA ZoeDepth-NK (Bhat et al., 2023) model. For surface normals estimation, we follow Prismer (Liu et al., 2023c) to use NLL-AngMF (Bae et al., 2021) that pre-trained on ScanNet (Dai et al., 2017). We decode video into frames to extract per-frame depth map/optical flow/surface normals. We set 3 fps, 1 fps, and 3 fps to decode SQA3D, MUSIC-AVQA, and NExT-QA videos respectively.

## A.3 MMQA PRE-TRAINING DETAILS

As discussed in Section 3.3, we conduct extra lightweight pre-training for MMQA module to obtain a good initialization. We follow audio-pertaining settings in X-InstructBLIP (Panagopoulou et al., 2023) with AudioCaps (Kim et al., 2019), but excluded caption data as our work is more focusing video reasoning. In this case, we obtained a QA-related subset from AudioCaps for MMQA-QA pre-training. Similarly, we utilized the 3D data released from 3D-LLM (Hong et al., 2023) and also took the QA format part for MMQA-3D pretraining. We pre-trained with $1e^{-5}$ learning rate and 1 epoch for efficient initialization.

## A.4 BASELINE MODEL IMPLEMENTATION

We conduct experiments with $4 \times 48$GB A6000 GPUs, we report baseline model training hyperparameters in Table 10. We follow hyperparameter settings that have been searched to yield the best performance in SeViLA (Yu et al., 2023a) with the same backbone model. To prompt LLM, we design different prompts for open-ended QA tasks (SQA3D, MUSIC-AVQA) and multi-choice QA (NExT-QA) following previous works (Yu et al., 2023a; Han et al., 2023). For open-ended QA, we let LLM generate responses without extra constraints and then compare the generated answers with ground-truth answers for accuracy calculation. We list prompts design for each dataset in Table 12. We utilize the same multimodal encoders, and multimodal information from the same estimator for fair comparison.

Table 11: CREMA fine-tuning hyperparameters.

| Dataset | Modality | # Frames | Batch Size per GPU | Learning Rate | Warmup | Epoch | Gradient Accumulation Step |
|---|---|---|---|---|---|---|---|
| SQA3D | P | 4 | 16 | 2e-4 | 1000 | 20 | 1 |
| | V | 4 | 16 | 2e-4 | 1000 | 20 | 1 |
| | V, P | 4 | 16 | 2e-4 | 1000 | 20 | 1 |
| | V, P, D | 4 | 16 | 2e-4 | 1000 | 20 | 1 |
| | V, P, D, N | 4 | 16 | 2e-4 | 1000 | 20 | 1 |
| MUSIC-AVQA | A | 4 | 24 | 2e-4 | 1000 | 20 | 1 |
| | V | 4 | 24 | 2e-4 | 1000 | 20 | 1 |
| | V, A | 4 | 24 | 2e-4 | 1000 | 20 | 1 |
| | V, A, F | 4 | 24 | 2e-4 | 1000 | 20 | 1 |
| | V, A, F, D, N | 4 | 16 | 2e-4 | 1000 | 20 | 1 |
| NExT-QA | V | 4 | 16 | 1e-4 | 1000 | 10 | 1 |
| | V, F | 4 | 16 | 1e-4 | 1000 | 10 | 1 |
| | V, D | 4 | 16 | 1e-4 | 1000 | 10 | 1 |
| | V, N | 4 | 16 | 1e-4 | 1000 | 10 | 1 |
| | V, F, D | 4 | 16 | 1e-4 | 1000 | 10 | 1 |
| | V, F, D, N | 4 | 8 | 1e-4 | 1000 | 10 | 2 |

Table 12: Prompt designs for each dataset.

| Dataset | LLM Prompt |
|---|---|
| SQA3D | *Based on the frames and 3D Model information, answer the question using a single word or phrase.* |
| MUSIC-AVQA | *Based on the frames and audio information, answer the question using a single word or phrase.* |
| VGGSound | *Based on the frames and audio information, answer the question using a single word or phrase.* |
| NExT-QA | *Considering the information presented in the frame, select the correct answer from the options* |
| PerceptionTest | *Considering the information presented in the frame, select the correct answer from the options* |
| TouchQA | *Based on the frames and touch map information, answer the question using a single word or phrase.* |
| ThermalQA | *Based on the frames and thermal heatmap information, answer the question using a single word or phrase.* |

## A.5 CREMA IMPLEMENTATION DETAILS

CREMA framework adopts BLIP-2 (Li et al., 2023b), an image-language model with 4.1B parameters and pre-trained on 129M images in total, including COCO (Lin et al., 2014), Visual Genome (Krishna et al., 2017), CC12M (Sharma et al., 2018), SBU (Ordonez et al., 2011), and 115M images from LAION400M (Schuhmann et al., 2021). See Appendix for details. we also report our CREMA framwork training hyperparameters in Table 11. The experiments are conducted on the same 4 × 48GB A6000 GPUs machine.

In the zero-shot setting, we conducted evaluations on SQA3D (video + point cloud) and MUSIC-AVQA (video + audio). Since these tests include only two modalities at a time, we bypassed the Self-Gated Multimodal Query Fusion module and directly concatenated the video tokens with the corresponding modality tokens. This ensures no parameter mismatch or interference and every modality is independent during inference.

Table 13: **Fine-tuning Results on VGGSound**. The numbers in the bracket represent results after excluding out-of-vocabulary results caused by formulating the classification as the open-ended QA.

| Method | Modality | Acc | Trainable Params. |
|---|---|---|---|
| CAV-MAE (Gong et al., 2022) | V | 47.0 | 324M |
| | V, A | 65.5 | 324M |
| Mirasol3B TTM (Piergiovanni et al., 2023) | V, A | 66.4 | 3B |
| Mirasol3B (Piergiovanni et al., 2023) | V, A | **69.8** | 3B |
| CREMA (ours) | V | 51.5 (53.1) | 4M |
| | V, A | 62.4 (67.0) | 9M |

Table 14: **Fine-tuning Results on PerceptionTest**.

| Method | Modality | Acc | Trainable Params. |
|---|---|---|---|
| BLIP-2 (Li et al., 2023b) | V | 67.1 | 108M |
| | V, F | 68.2 | 216M |
| | V, F, D | 67.9 | 324M |
| CREMA (ours) | V | 66.6 | 4M |
| | V, F | 68.2 | 8M |
| | V, F, D | **68.7** | 20M |

Table 15: **Comparison of Training Efficiency** on Video Question Answering (NExT-QA). Tested on the single A6000 GPU.

| Model | Modality | BatchSize | Training Memory |
|---|---|---|---|
| BLIP-2 (Li et al., 2023b) | V, F, D, N | 4 | 46.3GiB |
| CREMA | V, F, D, N | 4 | 18.6GiB |

# B EXTRA EXPERIMENTS

In this section, we provide additional experiments and analysis, including zero-shot per-task performance on MUSIC-AVQA (Section B.3), the impact of MMQA pre-training (Section B.5), the impact of LoRA rank (Section B.6), and more qualitative visualization (Section B.7).

Table 16: **Zero-shot Per-task Results of CREMA method on Audio-Video Question Answering (MUSIC-AVQA).** We report simple notations for each modality and question type: **V**: *Video RGB frames*, **A**: *Audio*, **Cnt.**: *Counting*, **Com.**: *Comparative*, **Loc.**: *Location*, **Ext.**: *Existential*, and **Tem.**: *Temporal*.

| Modality | Audio Question | | | Visual Question | | | Audio-Visual Question | | | | | | Avg. |
|---|---|---|---|---|---|---|---|---|---|---|---|---|---|
| | Cnt. | Com. | Avg. | Cnt. | Loc. | Avg. | Ext. | Loc. | Cnt. | Com. | Tem. | Avg. | |
| A | 50.4 | **53.2** | 51.0 | 29.1 | 18.5 | 23.9 | 39.9 | 13.0 | 27.1 | **49.6** | 4.3 | 29.1 | 31.0 |
| V | 73.4 | 51.2 | 68.6 | 51.4 | **44.4** | 48.0 | **76.4** | 43.5 | 38.7 | 47.1 | 26.3 | 47.7 | 51.0 |
| A,V | **75.5** | 51.6 | **70.4** | 55.5 | 42.6 | 49.2 | 76.2 | **44.2** | **45.1** | 48.2 | 26.3 | **49.5** | **52.6** |

## B.1 EXTRA RESULTS ON VGGSOUND AND PERCEPTIONTEST

As listed in Table 13 and Table 14, we report extra fine-tuning performance on VGGSound (Chen et al., 2020) and PerceptionTest (Pătrăucean et al., 2023). On VGGSound, we formulate the original audio-video classification task as an open-ended QA task. We calculate accuracy by directly matching the generated answer and the ground truth. Note that our original outputs have 2.9% (V) and 7.9% (V, A) predicted answers on test data that are out-of-vocabulary. We report both numbers with and without those out-of-vocabulary data in Section B.1. It shows that combined with audio (A) brings notable and consistent improvement. Our CREMA shows comparable performance with fewer trainable parameters compared with other methods. On PerceptionTest, our CREMA shows consistent improvement when adding new modalities while the baseline method struggles with this.

## B.2 COMPARISON OF TRAINING MEMORY

We further compare the training memory of our CREMA and baseline method in Table 15. Our method requires much less training memory when leveraging the same modality input and batch size. Thus, our method is more friendly to incorporate more modalities during the training in the view of computation resources.

Table 17: **Ablation of Modality-sequential Training** of CREMA.

| Method | Dataset | Modality | Accuracy | Trainable Params. |
|---|---|---|---|---|
| CREMA (Joint) | MUSIC-AVQA | V, A, F, D, N | 80.6 | 38M |
| | SQA3D | V, P, D, N | 52.7 | 38M |
| | NExT-QA | V, F, D, N | 73.0 | 28M |
| CREMA (Sequential) | MUSIC-AVQA | V, A, F, D, N | 81.7 (+1.1) | 38M |
| | SQA3D | V, P, D, N | 54.6 (+1.9) | 38M |
| | NExT-QA | V, F, D, N | 73.9 (+0.9) | 28M |

Table 18: **The impact of Modality-Specific LoRA Pre-training**. We report zero-shot performance.

| Dataset | Modality | w/o PT | w PT |
|---|---|---|---|
| MUSIC-AVQA (Avg.) | A | 28.1 | 31.0 |
| SQA3D (Avg.) | P | 36.1 | 37.3 |

## B.3 EXTRA ZERO-SHOT RESULTS ON MUSIC-AVQA

As listed in Table 16, we report extra zero-shot performance on MUSIC-AVQA by more fine-grained task/question types. It shows that video (V) combined with audio (A) brings notable and consistent improvement across most question types, highlighting the compositional video reasoning ability of our proposed CREMA.

## B.4 THE EFFECT OF MODALITY-SEQUENTIAL TRAINING

We validate the effectiveness of the modality-sequential training of our approach in Table 17. We report the fine-tuning performance on three datasets: MUSIC-AVQA, SQA3D, and NExT-QA. As shown, our sequential training mechanism effectively improves multimodal video reasoning capabilities without additional architectures or the computationally and memory-intensive gradient modification strategy. This advantage is distinguishable from MLA (Zhang et al., 2024), a multimodal representation learning method that sequentially trains a shared classification head on different modalities. Since MLA uses a unified trainable module (*i.e.*, the shared classifier head) to train multimodal data, it employs the orthogonal gradient projection technique on different sensory inputs to mitigate the forgetting of previously learned modality information. In contrast, our model leverages substantial capacity to capture crucial information from various modalities through lightweight, yet effective modality-specific multi-query adapter (MMQA) modules, avoiding modality forgetting. Furthermore, we propose a modality-adaptive dynamic early exit strategy, enabling our model to converge quickly and mitigate over-convergence issues in different modality data by utilizing the gradients of lightweight modality-specific modules.

## B.5 THE IMPACT OF MMQA PRE-TRAINING.

As listed in Table 18, we demonstrate the impact of MMQA module pre-training on SQA3D and MUSIC-AVQA datasets. It shows that such an efficient MMQA pre-training brings a significant boost (+1.2% on SQA3D with 3D point could (P), +2.9% on MUSIC-AVQA with audio (A)) to the zero-shot performance for each single modality. It demonstrates the effectiveness of our MMQA and the pre-training process.

Table 19: **The impact of the rank** $r$ of modality-adaptive LoRA module in CREMA.

| #Rank | Music-AVQA (A) | SQA3D (P) | Trainable Params. (A) / (P) |
|---|---|---|---|
| 32 | **69.4** | 47.0 | 3.8 M / 2.7 M |
| 64 | 68.0 | **47.3** | 5.0 M / 3.9 M |
| 128 | 67.7 | 46.3 | 7.4 M / 6.3 M |

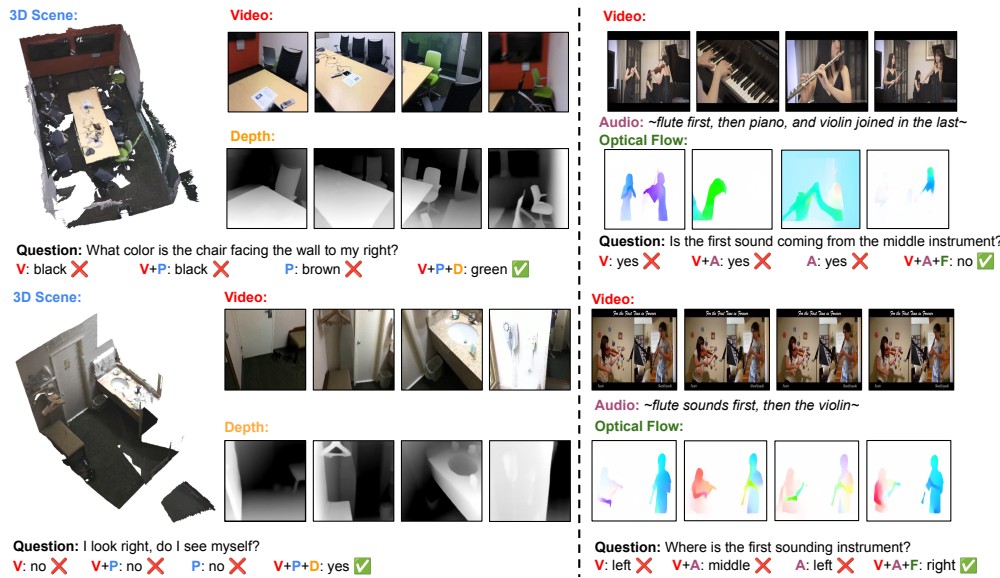

Figure 3: **Qualitative examples for multimodal compositional video reasoning** from SQA3D (Left) and MUSIC-AVQA (Right). The correct predictions are marked by green check marks.

## B.6 RANK OF LoRA MODULE.

We investigate the impact of the rank $r$ of our modality-adaptive LoRA modules in CREMA. As shown in Table 19, adjusting the rank size within a reasonable range brings an insignificant change in the number of trainable parameters (e.g., $\pm 1 \sim 2$ M). We also find that a larger rank size does not guarantee improved performance. For MUSIC-AVQA, we fine-tuned the model on audio inputs (A), and it always outperforms the best-performing baseline (AVQA, 64.2%) by a significant margin. CREMA trained on 3d point cloud data performs similarly to the rank size of $32$ and $64$ on SQA3D. We set $r = 64$ as the default for all experiments, showing the robustness of our MMQA module in selecting $r$ over various modality inputs and evaluation tasks. But we believe that our CREMA method with proper $r$ can further improve its reasoning ability.

## B.7 QUALITATIVE ANALYSIS

Beyond the numerical comparison of the effect integrating different sets of modalities for our CREMA method, we investigate our model's generated responses according to different types of input examples. In Figure 3 Left, CREMA with 3D point cloud inputs (P) fails to find the chair and respond to the color of the wall, brown, as its 2D scene image features are incorporated in 3D point cloud features. CREMA with Video (V) and *V, P* also predict incorrect chair color, black. However, with the assistance of depth information, the method can capture objects accurately and also find the designated chair. Similarly, in Figure 3 Right, optical flow inputs help to find musicians with their poses playing instruments, so our CREMA method can tell the middle instrument is not being played at the beginning, but from the left. In Figure 4, we present additional visual examples from SQA3D and MUSIC-AVQA, demonstrating how the integration of multiple input modalities enhances model predictions. For instance, depth maps in the top-left example reveal the distance of objects, enabling the model to discern that the clock is closer than the pillow. Similarly, on the middle left, depth maps indicate an open door through depth of field analysis, aiding in question answering. In the MUSIC-AVQA examples on the right, the optical flow captures motion, which is essential for deducing which instrument is being played. Specifically, the bottom right illustration shows that the initial static behavior of the left people implies that the right instrument is not played initially. This evidence highlights the benefit of incorporating diverse modalities for improved model reasoning ability.

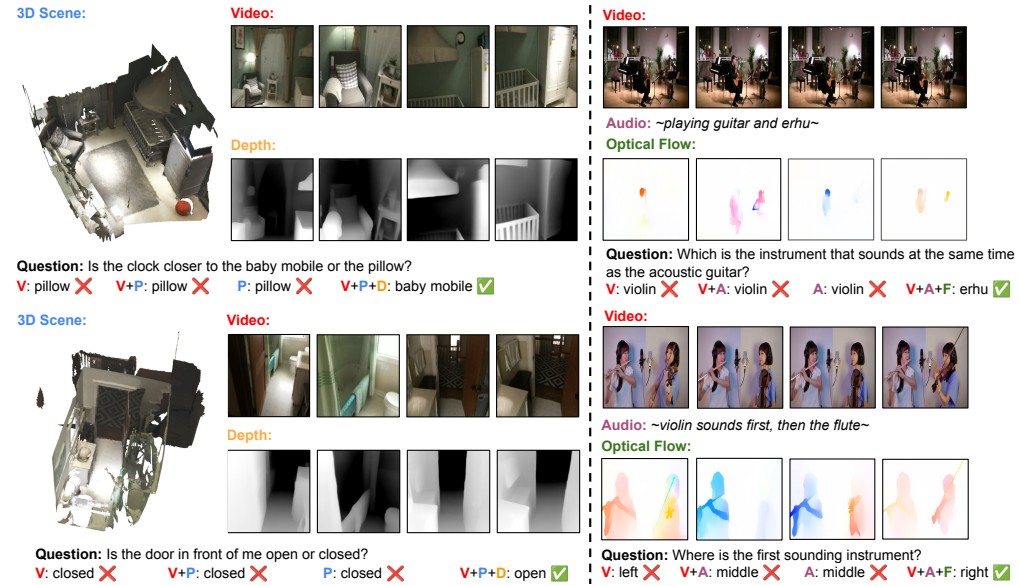

Figure 4: Qualitative examples for multimodal compositional video reasoning from SQA3D (Left) and MUSIC-AVQA (Right). The correct predictions are marked by green checks.

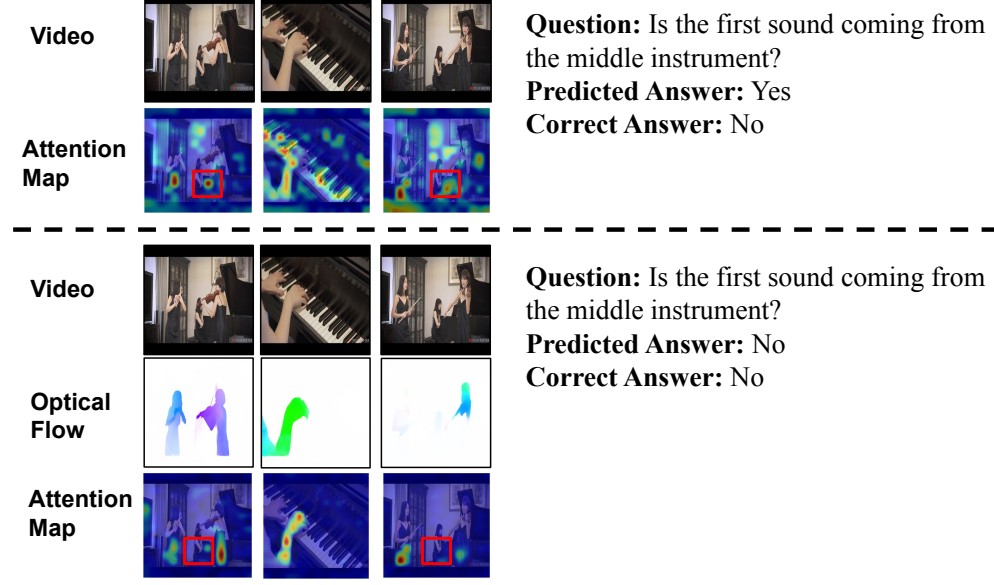

Figure 5: Visualization on attention map under different modality combinations. **Top**: with audio and video. **Bottom**: with audio, optical flow, and video. We omit audio for simplicity. We highlight attention regions that may affect model prediction with red boxes.

## B.8 EXTRA DISCUSSIONS ON MODEL DESIGN.

**Prioritizing different modalities.** We conducted additional experiments comparing different prioritization strategies, including prioritizing other modalities and treating all modalities equally (i.e., no fusion, directly concatenating tokens) in Table 20. It shows that prioritizing video achieves the best performance, and prioritizing other modalities or treating all modalities equally results in lower performance, validating the effectiveness of our design.

**Single FC layer Design.** we conducted additional experiments comparing the FC layer to a lighter-weight alternative, interpolation to project dimension-mismatched features between multimodal encoder and multimodal Q-Former in Table 21. The results show that the FC layer improves

| Setting | NExT-QA |
|---------|---------|
| Major: V + Supportive: D,F,N | 73.9 |
| Major: D + Supportive: V,F,N | 62.1 |
| No Prioritizing: V, D, F, N | 73.5 |

Table 20: Ablation on prioritizing different modalities.

| Connector | # Parameters | MUSIC-AVQA |
|-----------|--------------|------------|
| One-layer FC | 0.3M | 79.4 |
| Interpolation | 0M | 78.9 |

Table 21: Ablation on the projection layer between multimodal encoder and multimodal Q-Former.

performance while adding a negligible number of parameters. We attribute this improvement to the FC layer's ability to provide a more dynamic and learnable projection between multimodal encoders and the Multimodal QFormer, which better aligns features compared to interpolation.

**Query Token Length.** We conducted experiments analyzing the impact of query token length on performance, trainable parameters, and computational cost in Table 22. We find increasing the number of query tokens indeed improves accuracy as more fine-grained features are captured. However, it also leads to increasing computational costs (GFLOPs). We find that 32 query tokens per frame strike a good balance between performance and efficiency. This design aligns with BLIP-2, ensuring strong performance without excessive computational overhead.

## C  LIMITATION & BROADER IMPACTS

**Limitations.** Our CREMA is mainly based on BLIP-2 structure which was originally pre-trained only on image-language data. This may lead to limited temporal modeling/knowledge in the pre-trained model. It might not handle well fine-grained temporal events (e.g., open the door vs. close the door). Moreover, a potential limitation could be incorporating multimodal inputs such as depth and flow, which requires computing these modalities and introduces additional overhead.

**Broader Impacts.** The CREMA framework leverages a pre-trained vision-language model backbone with the proposed adapter modules to integrate multiple modality inputs through a universal framework. Similar to most works leveraging pre-trained vision-language models, this might occasionally yield unexpected or inappropriate responses, potentially reflecting societal biases related to gender, race, or sexuality. More studies of vision-language models are needed to evaluate and mitigate these negative biases, and toxic output.

## D  LICENSE

We will make our code and models publicly accessible. We use standard licenses from the community and provide the following links to the licenses for the datasets, codes, and models that we used in this paper. For further information, please refer to the specific link.

**SQA3D:** Apache

| Modalities | # Quey Token | NExT-QA | Trainable Param. | GFlops |
|------------|--------------|---------|------------------|--------|
| V | 16 | 70.8 | ∼4M | 1.0K |
| V | 32 | 71.6 | ∼4M | 1.3 K |
| V | 64 | 72.0 | ∼4M | 2.1 K |
| V,F | 16 | 71.8 | ∼8M | 1.4K |
| V,F | 32 | 72.4 | ∼8M | 2.2K |
| V,F | 64 | 72.9 | ∼8M | 6.2K |

Table 22: Ablation on the number of query tokens.

**MUSIC-AVQA:** MIT

**NExT-QA:** MIT

**AudioCaps:** MIT

**3D-LLM:** MIT

**LAVIS:** BSD 3-Clause

**Touch-and-Go:** CC BY

**Thermal-IM:** BSD 3-Clause

**PerceptionTest:** Apache

**VGGSound:** CC BY

**PyTorch:** BSD-style

**Huggingface Transformers:** Apache

**Torchvision:** BSD 3-Clause

