# OpenReview forum: "CREMA: Generalizable and Efficient Video-Language Reasoning via Multimodal Modular Fusion"
_ICLR.cc/2025/Conference — ICLR 2025 Poster_

### Official Review · Reviewer_bc5T · 2024-10-26

**Soundness:** 2
**Presentation:** 3
**Contribution:** 2
**Rating:** 6
**Confidence:** 4

**Summary:**

This work presents a multi-modal LLM pipeline CREMA to joint learning from different modalities: visual, depth map, optical flow, audio etc that are synchronized with a video input. Built on top of existing multimodal encoders and LLM, it proposes modal-specific queries and a modality fusion module to incorporate inputs from different modalities while keeping a low trainable parameter scale. The model is evaluated on tasks require multimodal inputs: audio-video QA, 3D situated QA, touchQA/thermal QA etc. It outperforms existing methods that are using the multimodal inputs such as OneLLM and 3D-LLM.

**Strengths:**

-	The proposed model is a general framework for language QA based on multimodal video inputs. It achieves impressive performance on a wide range of tasks: audio-video QA, 3D situated QA, touchQA/thermal QA etc.
-	Some ablation studies are conducted for the choice and early exit strategy and modality fusion method (Table 7&8).

**Weaknesses:**

-	The performance of the proposed model seems to be dependent on the used multimodal encoders (ZoeDepth, Unimatch, NLL-AngMF to estimate depth, flow, normal, BEATs to encode audio, and ViT-G to encode visual). The comparison to existing methods might be unfair due to different encoders are used. More explanations are needed to verify this.
-	The overall novelty is limited. The proposed model-specific queries and modality fusion module are subtle technical changes that does not bear a strong novelty.

**Questions:**

-  Motivation behind adaptive early exit is not clear. The equation used on line 284 says if the gradient for a modality is larger than a threshold, then it will exit training. Shouldn’t it be smaller than a threshold since the gradient scale will be small after convergence?
- Why using a sigmoid function in the fusion module in equation (3)? Seems it only does a scaling to the original q^{\bar}_\V which may not be necessary

---

> ### Author Response · Authors · 2024-11-20
> **Official Comment by Authors (1)**
>
> Thank you for your review and constructive comments. During the rebuttal period, we have made every effort to address your concerns. The detailed responses are below:
>
> > **W1**: The performance of the proposed model seems to be dependent on the used multimodal encoders (ZoeDepth, Unimatch, NLL-AngMF to estimate depth, flow, normal, BEATs to encode audio, and ViT-G to encode visual). The comparison to existing methods might be unfair due to different encoders are used. More explanations are needed to verify this.
>
> Thank you for your valuable feedback. We would like to clarify that we used **the same multimodal encoders** for our main baselines (3D-LLM, BLIP-2, X-BLIP) as we did for our proposed CREMA. Specifically, we adopted **the same multimodal information**—such as depth, optical flow, and normals—obtained from **the same estimation models** (ZoeDepth, Unimatch, NLL-AngMF, etc.) for all models in our experiments. We used **the same visual encoder (ViT-G) and audio encoder (BEATs)** across comparisons.
>
> By keeping the input data and encoders consistent among all models, the only differences lie in the model design and training strategy. This ensures that any performance variations are due to our proposed methods rather than differences in the encoders.
> Therefore, our comparisons are fair and valid, demonstrating the strong effectiveness of the CREMA framework.
>
> We **have added more clarification  (Line 964-965)** in our revision to address this concern.
>
> ---
> > **W2**: The overall novelty is limited. The proposed model-specific queries and modality fusion module are subtle technical changes that do not bear a strong novelty.
>
> Thank you for your feedback! We respectfully clarify the key novelties/contributions of the CREMA framework:
>
> - **Novel Framework Design and Unique Training Strategy**: CREMA introduces a new model design with components like the multimodal Q-former, Modality-Specific Multi-Query Adapter, and Self-Gated Fusion. It also incorporates a novel modality-sequential training strategy tailored for efficient optimization across multiple modalities.
>
> - **Generalizability and Efficiency**: CREMA is the first highly efficient and generalizable modality-extensible learning framework for video-language reasoning. It enables seamless integration of video, language, and additional modalities with minimal computational resources, while delivering consistently strong performance across diverse benchmarks (validated on 7 datasets and 9 modalities).
>
> - **Strong Performance with Less Resource Demand**: CREMA outperforms strong multimodal models (e.g., BLIP-2, 3D-LLM, OneLLM, X-BLIP) with better scalability on modalities and significantly lower resource requirements.
>
> Additionally, we kindly note that **other reviewers have recognized CREMA’s contributions/novelties**, we quote their comments as follows:
>
> - Reviewer ```dN8u```: It's the first time I've seen the use of gradients to determine whether to exit early, and it is also the first method to apply early stopping by modality. Therefore, I acknowledge the paper's innovative approach.
> - Reviewer ```5ndk```: The proposed fusion approach with Q-former (architecture) and modality-sequential training (training recipe) are both reasonable and look simple for other researchers to follow.
> - Reviewer ```wvMi```: This paper introduces CREMA, a novel, parameter-efficient framework that enables the seamless addition of new modalities without altering the core architecture—a significant advantage over existing models.
>
> We acknowledge and understand that perspectives on novelty could be varied/subjective, but we hope this clarification and the supporting reviewer comments help underline the unique contributions of our work.
>
> We are open to incorporating any further suggestions and kindly request the reviewer to reconsider the rating. If there are additional concerns or questions, we would be glad to provide further clarification. Thank you again for your time and review.
>
> ---
> > **Q1**: Motivation behind adaptive early exit is not clear.  The equation used on line 284 says if the gradient for a modality is larger than a threshold, then it will exit training. Shouldn’t it be smaller than a threshold since the gradient scale will be small after convergence?
>
> The original expression is correct. As the reviewer noted, the gradient scale diminishes over time, indicating that the average mean gradient across all prior epochs gradually decreases. This loosely satisfies the condition average(\bar{g[:j]}) >= average(\bar{g[:j+1]}), though not strictly, due to the stochastic nature of the optimization process.
>
> We interpret this behavior as an indication that the modality information in CREMA has converged when the average mean gradient across all previous epochs (with temperature \tau) has decreased sufficiently to fall below the mean gradient of the most recent epoch (j+1).

---

> > ### Author Response · Authors · 2024-11-20
> > **Official Comment by Authors (2)**
> >
> > > **Q2**: Why using a sigmoid function in the fusion module in equation (3). Seems it only does a scaling to the original q^{\bar}_\V which may not be necessary
> >
> > As explained in Lines 235-236, the sigmoid function acts as a gating mechanism in our self-gated operation. By applying the sigmoid function to q^{\bar}_\V, we gate the feature with itself without introducing additional parameters.
> > The sigmoid function outputs values between 0 and 1, allowing the model to scale the feature dynamically. This gating mechanism serves to amplify or suppress parts of the feature, enabling the model to learn and focus on the most useful information from the diverse assistant modalities.
> >
> > This is particularly beneficial when handling multiple modalities, as it helps in distinguishing and extracting relevant signals. To further support the effectiveness claim about the sigmoid module, in this rebuttal, we also provide extra ablation studies as follows. It demonstrates this self-gated operation with sigmoid function can help performance.
> >
> > Setting | NExT-QA Acc.
> > |-|-|
> > V, F (w sigmoid) | 72.4
> > V, F (w/o sigmoid) |  72.0

---

> > > ### Comment · Reviewer_bc5T · 2024-11-22
> > >
> > > Thanks for the detailed response. My concerns are mostly addressed, though I still have some concerns on the novelty part. I believe the comprehensive study with the multi-modality information for multiple tasks carries the major value of this work. Will raise my score if other reviewers do not express additional concerns.

---

> > > > ### Author Response · Authors · 2024-11-23
> > > > **Response to Reviewer bc5T**
> > > >
> > > > Dear Reviewer bc5T,
> > > >
> > > > Thank you for your response to our rebuttal. We are glad to know that most of your concerns have been addressed!
> > > >
> > > > We agree that the "comprehensive study with multi-modality information for multiple tasks" is a key contribution of our work. To emphasize this, **we have explicitly highlighted it in the revised version (Line 129-130)**.
> > > >
> > > > We sincerely appreciate your willingness to raise your score/rating. If there are any further comments or discussions needed, we would be happy to provide additional clarification to strengthen our paper. Thank you again for your time and valuable input!

---

> > > > > ### Author Response · Authors · 2024-12-01
> > > > > **Friendly Follow-Up on CREMA. We have less than two days left in the discussion period.**
> > > > >
> > > > > Dear bc5T,
> > > > >
> > > > > Thank you for your thoughtful feedback and for taking the time to review our paper. We wanted to kindly follow up and share that other active reviewers have expressed positive feedback with no additional concerns after our rebuttal:
> > > > >
> > > > > > Reviewer ```dN8u```: *The experimental results are very meaningful; I have raised my score to 8.*
> > > > >
> > > > > > Reviewer ```5ndk```: *Thank you for the additional clarifications! I don't have further questions.*
> > > > >
> > > > > > Reviewer ```wvMi```: *I believe you have comprehensively addressed my concerns, and the updates significantly strengthen the paper. I will increase the score to 8.*
> > > > >
> > > > > We sincerely appreciate your insights and remain available to address any remaining concerns. We kindly request the reviewer to reconsider your score/rating. Thank you again for your time and valuable feedback.
> > > > >
> > > > > Best regards,
> > > > >
> > > > > Authors

---

### Official Review · Reviewer_wvMi · 2024-10-26

**Soundness:** 3
**Presentation:** 3
**Contribution:** 3
**Rating:** 8
**Confidence:** 5

**Summary:**

The paper proposes CREMA, a flexible and efficient framework for video-language reasoning that incorporates multiple modalities, including optical flow, audio, thermal maps, and 3D point clouds. CREMA addresses the limitations of current multimodal models that require extensive parameters and fixed modality inputs by introducing a modular, parameter-efficient design. This framework allows seamless integration of new modalities while reducing computational costs, validated by superior performance on seven diverse reasoning tasks compared to baseline models.

**Strengths:**

This paper introduces CREMA, a novel, parameter-efficient framework that enables the seamless addition of new modalities without altering the core architecture—a significant advantage over existing models like BLIP-2 and SeViLA, which rely on fixed modality inputs and require extensive parameters. CREMA effectively integrates diverse modalities, such as 3D, thermal, and audio data, by projecting them into a unified representation space interpretable by the model for reasoning.

Key architectural innovations, including self-gated multimodal query fusion and sequential modality training, bring practical improvements to multimodal reasoning tasks, particularly in video-language applications. CREMA demonstrates broad applicability and efficiency across seven video-language reasoning tasks, achieving notable accuracy gains in VideoQA and 3D reasoning. Through reductions of over 90% in parameter requirements and optimizations like modality-sequential training and adaptive early exit, CREMA marks a significant advancement in multimodal reasoning, validated through extensive fine-tuning and zero-shot evaluations.

**Weaknesses:**

●  Prioritizing certain modalities as primary lacks quantitative backing, which could benefit from sensitivity analysis to validate this design choice across diverse tasks.

●  The Q-Former generates fixed-length tokens for each modality to extract the most informative features and remove irrelevant information. However, this fixed-length constraint could risk omitting valuable details, particularly in modalities with high information density.

●  The decomposition of back-propagation by modality, while efficient, may limit the model’s ability to fully capture interactions between modalities, impacting the quality of multimodal reasoning.

**Questions:**

●Q1: In Line 177, the paper states that the Q-Former "extracts the most informative features from the input modality and removes any irrelevant information" by generating fixed-length tokens. However, the fixed-length constraint may risk omitting crucial details, particularly for modalities rich in information. To substantiate the claim of extracting only the most informative features, it would be beneficial to include empirical evidence or an ablation study comparing different token lengths and their impact on performance across modalities.

●Q2: In Lines 194-195, the authors mention adding a fully connected layer (shown as a dashed box in Figure 1) for each data type when dimension misalignment occurs. Could you clarify why a fully connected layer was chosen over a potentially lighter-weight approach like interpolation? A fully connected layer seems more computationally intensive, so I am curious about the specific advantages it offers in this context. To clarify the advantages of this design choice, it would be helpful for the authors to provide a brief comparison, perhaps in terms of computational cost and performance, with lighter-weight options such as interpolation.

●Q3: In Line 233, the authors select video queries as the "major" modality, with other modalities as "supportive," explaining this choice as mirroring human perception in video reasoning tasks. Could you clarify the rationale behind prioritizing video in this way? Additionally, was a sensitivity analysis conducted to verify the impact of this design choice? I am curious whether this prioritization consistently benefits performance across tasks or if certain scenarios might require a different modality emphasis. To support this prioritization, the authors could consider presenting results from ablation studies or sensitivity analyses across various tasks and modality combinations, demonstrating whether prioritizing video consistently enhances performance or if other scenarios might benefit from different modality emphasis.

● Q4: In Line 259, the authors describe decomposing the back-propagation process for different modalities in each iteration. Could this approach limit the model’s ability to capture interactions between modalities, which is critical for vision-language tasks? It seems more like a trade-off for efficiency rather than a true remedy, as mentioned. This decomposition may prevent the model from fully learning cross-modal interactions and effectively fusing information across modalities. Could you clarify this design choice and its potential impact on performance?

● Q5: In Line 456, the paper mentions achieving a 'regularization effect on the model through parameter-efficient updates.' Could you elaborate on the specific mechanisms or components within CREMA that contribute to this regularization effect? Additionally, how does this approach enhance model generalization across various modalities?

● Q6: Could CREMA also accommodate remote sensing imagery as an input modality? Remote sensing images, captured from satellites or drones, provide detailed information on Earth’s surface across multiple spectral bands. If CREMA can process this type of data, would specific adaptations be needed to handle its unique spatial and spectral characteristics?

---

> ### Author Response · Authors · 2024-11-20
> **Official Comment by Authors (1)**
>
> Thank you for your review and constructive comments. During the rebuttal period, we have made every effort to address your concerns. The detailed responses are below:
>
> > **W1**: Prioritizing certain modalities as primary lacks quantitative backing, which could benefit from sensitivity analysis to validate this design choice across diverse tasks.
>
> We have conducted extra experiments to compare different prioritizing modalities in this rebuttal. A quick clarification for our motivation behind this design includes two folds:
> - **CREMA focuses on video-language reasoning tasks**. All 7 benchmarks/datasets inherently rely heavily on video information.
> - **CREMA is built on vision-language models** like BLIP-2 (Tables 1-3) and VideoChat2 (Table 6). Video modality has the smallest domain gap to the backbone model.
>
>  Please refer to **Q3** for more numbers and discussion. Thank you!
>
> ---
> > **W2**: The Q-Former generates fixed-length tokens for each modality to extract the most informative features and remove irrelevant information. However, this fixed-length constraint could risk omitting valuable details, particularly in modalities with high information density.
>
> We conduct extra experiments on different token lengths in this rebuttal. A quick conclusion: Through experiments, we find our query token number design achieves **the best performance and training cost balance**.
>
> Please refer to **Q1** for more numbers and discussion. Thank you!
>
> ---
> > **W3**: The decomposition of back-propagation by modality, while efficient, may limit the model’s ability to fully capture interactions between modalities, impacting the quality of multimodal reasoning.
>
> We believe we do not suffer from some sub-optimal interactions among modalities during our sequentially modality training procoess. It is also supported by experiments comparison between joint training and sequential training in **Table 17** (Appendix). Please refer to **Q4** for more clarification.
>
> ---
> > **Q1**: In Line 177, the paper states that the Q-Former "extracts the most informative features from the input modality and removes any irrelevant information" by generating fixed-length tokens. However, the fixed-length constraint may risk omitting crucial details, particularly for modalities rich in information. To substantiate the claim of extracting only the most informative features, it would be beneficial to include empirical evidence or an ablation study comparing different token lengths and their impact on performance across modalities.
>
> Thank you for your insightful suggestion. To address your concern, we conducted experiments analyzing the impact of query token length on performance, trainable parameters, and computational cost. Below are the results from evaluations on the NExT-QA dataset:
>
> Modalities | # Quey Token | NExT-QA Acc. | # trainable param. | GFlops
> -|-|-|-|-|
> V         |          16         |     70.8   |    ~4M   | 1.0K
> V           |         32         |     71.6   |     ~4M  |  1.3 K
> V            |        64         |     72.0  |  ~4M   |  2.1 K
> V,F          |       16         |    71.8   |  ~8M  | 1.4K
> V,F        |      32         |     72.4 | ~8M |   2.2K
> V,F        |      64         |    72.9 | ~8M |    6.2K
>
> We find increasing the number of query tokens indeed improves accuracy as more fine-grained features are captured. However, it also leads to increasing computational costs (GFLOPs). We find that 32 query tokens per frame strike a good balance between performance and efficiency. This design aligns with BLIP-2, ensuring strong performance without excessive computational overhead.
>
> The results substantiate our claim that the Q-Former extracts the most informative features with a fixed-length token representation. We will include these findings in our revision to provide clear empirical evidence supporting our design choice. Thank you again for your valuable feedback!

---

> > ### Author Response · Authors · 2024-11-20
> > **Official Comment by Authors (2)**
> >
> > > **Q2**: In Lines 194-195, the authors mention adding a fully connected layer (shown as a dashed box in Figure 1) for each data type when dimension misalignment occurs. Could you clarify why a fully connected layer was chosen over a potentially lighter-weight approach like interpolation? A fully connected layer seems more computationally intensive, so I am curious about the specific advantages it offers in this context. To clarify the advantages of this design choice, it would be helpful for the authors to provide a brief comparison, perhaps in terms of computational cost and performance, with lighter-weight options such as interpolation.
> >
> > We opted for a single FC layer because it is a widely adopted and simple approach in multimodal LLM works for feature dimension mismatchings, such as LLaVA and BLIP-2. Additionally, **it introduces minimal trainable parameters**—for example, for the audio modality, the FC layer adds only **~0.3M** parameters (512x768).
> >
> > To further evaluate this design choice, we conducted additional experiments comparing the FC layer to a lighter-weight alternative, interpolation. Below are the results:
> >
> > Connector | #Parameters | Music-AVQA Acc. (with Video and Audio)
> > |-|-|-|
> > One-layer FC | 0.3M | 79.4
> > Interpolation | 0M | 78.9
> >
> > The results show that the FC layer improves performance while adding a negligible number of parameters. We attribute this improvement to the FC layer’s ability to provide a more dynamic and learnable projection between multimodal encoders and the Multimodal QFormer, which better aligns features compared to interpolation.
> >
> > We will include this comparison in our revision to clarify the advantages of the FC layer in this context. Thank you again for your valuable suggestion!
> >
> > ---
> >
> > > **Q3**: In Line 233, the authors select video queries as the "major" modality, with other modalities as "supportive," explaining this choice as mirroring human perception in video reasoning tasks. Could you clarify the rationale behind prioritizing video in this way? Additionally, was a sensitivity analysis conducted to verify the impact of this design choice? I am curious whether this prioritization consistently benefits performance across tasks or if certain scenarios might require a different modality emphasis. To support this prioritization, the authors could consider presenting results from ablation studies or sensitivity analyses across various tasks and modality combinations, demonstrating whether prioritizing video consistently enhances performance or if other scenarios might benefit from different modality emphasis.
> >
> > Thank you for raising this question. Beyond the reasoning provided in Line 233, the primary rationale for prioritizing video as the "major" modality is rooted in the nature of our target tasks and the architecture of our framework:
> >
> >
> > - **Task Definition**: All 7 datasets/benchmarks we evaluate are video-language reasoning/QA tasks. These tasks inherently rely heavily on video information, as it provide the richest context for video understanding.
> > - **Model Backbone**: CREMA is built on vision-language models like BLIP-2 (Tables 1-3) and VideoChat2 (Table 6), which are pre-trained on massive visual-language data. Prioritizing the visual modality (video) aligns with the strengths of these pre-trained models, minimizing domain gaps and maximizing their effectiveness.
> >
> > To further validate this design choice, we conducted additional experiments comparing different prioritization strategies, including prioritizing other modalities and treating all modalities equally (i.e., no fusion, directly concatenating tokens). Below are the results:
> >
> > Setting | NExT-QA Acc.
> > -|-
> > Major: V + Supportive: D,F,N |  73.9
> > Major: D + Supportive: V,F,N  |  62.1
> > No Prioritizing: V, D, F, N  | 73.5
> >
> > It shows that prioritizing video achieves the best performance, and prioritizing other modalities or treating all modalities equally results in lower performance, validating the effectiveness of our design.
> >
> > We also acknowledge that prioritizing other modalities might be better in other domain-specific tasks (e.g., audio in audio classification or point clouds in 3D scene navigation). However, such tasks fall outside the scope of this work, as CREMA focuses on video-language reasoning.
> >
> > We hope this clarification and the additional results provide a clear rationale for our design choice. Thank you for your thoughtful suggestion!

---

> > > ### Author Response · Authors · 2024-11-20
> > > **Official Comment by Authors (3)**
> > >
> > > > **Q4**: In Line 259, the authors describe decomposing the back-propagation process for different modalities in each iteration. Could this approach limit the model’s ability to capture interactions between modalities, which is critical for vision-language tasks? It seems more like a trade-off for efficiency rather than a true remedy, as mentioned. This decomposition may prevent the model from fully learning cross-modal interactions and effectively fusing information across modalities. Could you clarify this design choice and its potential impact on performance?
> > >
> > > Thank you for your insightful question. We understand the concern regarding potential limitations in capturing cross-modal interactions during our sequential modality training process. However, we would like to clarify that our approach still ensures robust multimodal interactions.
> > >
> > > Specifically, while weights for each modality are updated sequentially, the final loss computation always involves all modalities. This ensures that strong cross-modal interactions are captured and optimized. By freezing weights of other modalities during updates for each specific modality within an iteration, we prevent negative interference and mitigate the risk of sub-optimal learning. This design allows us to intelligently balance efficient training with effective multimodal integration. The effectiveness of our method is also supported by the performance comparison between joint optimization and sequential training in **Table 17** (Appendix).
> > >
> > > Additionally, this approach offers flexibility, enabling seamless incorporation of new modalities, which is challenging for other MLLM frameworks.
> > >
> > > ---
> > >
> > > > **Q5**: In Line 456, the paper mentions achieving a 'regularization effect on the model through parameter-efficient updates.' Could you elaborate on the specific mechanisms or components within CREMA that contribute to this regularization effect? Additionally, how does this approach enhance model generalization across various modalities?
> > >
> > > Thank you for your question. The regularization effect of parameter-efficient updates is supported by both theory [1] and empirical evidence [2,3]. Specifically, updating only lightweight modules (e.g., LoRA) while keeping the large pre-trained backbone intact acts as a form of implicit regularization [3]. Recent work [3] also shows that LoRA fine-tuning reduces forgetting compared to traditional techniques like dropout or weight decay, while also maintaining diversity in model outputs.
> > >
> > > CREMA enhances model generalization across various modalities by leveraging principles inspired by sparse Mixture-of-Experts (MoE) designs, which are well-known for their strong generalization capabilities [4]. Similarly, our proposed framework, CREMA, approximates the generalization ability of sparse MoEs through two key innovations: Modality-Specific Multi-Query Adapter (MMQA) and Self-Gated Multimodal Query Fusion.
> > >
> > > MMQA is specifically designed to efficiently process modality-specific information, ensuring the preservation and effective utilization of unique features inherent to each modality, which enhances the model’s adaptability across diverse tasks. The proposed multimodal fusion design dynamically integrates outputs from different modalities, using a self-gating architecture to prioritize and fuse multimodal information based on task-specific requirements, mimicking the sparse selection router design in Sparse MoE architectures. This fusion approach enables the model to effectively balance and combine diverse inputs, further boosting its generalization ability.
> > >
> > > [1] Fu et al, On the Effectiveness of Parameter-Efficient Fine-Tuning, AAAI 2023.
> > > [2] Sun et al., Exploring the impact of low-rank adaptation on the performance, efficiency, and regularization of RLHF, arXiv 2309.09055.
> > > [3]Biderman et al., LoRA Learns Less and Forgets Less, TMLR 2024.
> > > [4] Li et al., Sparse Mixture-of-Experts are Domain Generalizable Learners, ICLR 2023 Oral presentation (notable-top-5%)

---

> > > > ### Author Response · Authors · 2024-11-20
> > > > **Official Comment by Authors (4)**
> > > >
> > > > > **Q6**: Could CREMA also accommodate remote sensing imagery as an input modality? Remote sensing images, captured from satellites or drones, provide detailed information on Earth’s surface across multiple spectral bands. If CREMA can process this type of data, would specific adaptations be needed to handle its unique spatial and spectral characteristics?
> > > >
> > > > Thank you for raising this future direction. While we are not experts in remote sensing imagery processing, we believe CREMA’s general framework can be extended to accommodate remote sensing data due to its flexible and efficient design for multimodal learning.
> > > >
> > > > Here are some thoughts from authors on how CREMA could adapt to handle remote sensing imagery and its unique spatial and spectral characteristics:
> > > >
> > > > - **Appropriate Multimodal Encoders**: Adapting CREMA would involve selecting or fine-tuning specialized encoders, such as spectral feature extractors (e.g., CNN-based models for hyperspectral data) or other transformer-based encoders for high-resolution spatial features. These encoders could replace or complement existing modules like the video or audio encoders in CREMA.
> > > > - **Stronger VLM Backbones**: Using advanced VLM backbones, such as Qwen2-VL, would enhance the framework’s ability to process and reason over remote sensing data when integrated with other modalities.
> > > >
> > > > We also note that CREMA’s modality-adaptive training and early exit strategies are particularly suitable for handling diverse input types including spectral bands or spatial data. Reviewer ```dN8u``` has also recognized this aspect, noting that the modality-adaptive early exit strategy has broad application potential.
> > > >
> > > > While specific adaptations would be required, we believe the CREMA framework provides a solid foundation for easily exploring remote sensing imagery as a new modality.

---

> > > > > ### Comment · Reviewer_wvMi · 2024-11-22
> > > > >
> > > > > Thank you for the discussion about incorporating additional modality data, such as remote sensing imagery, into CREMA. Your explanation regarding the framework's flexibility and its potential to adapt to the unique characteristics of remote sensing data is clear and well-considered. I am satisfied with your answer to Q6.

---

> > > > ### Comment · Reviewer_wvMi · 2024-11-22
> > > >
> > > > For Q4, the authors explain that while weights for each modality are updated sequentially during training, the final loss computation includes all modalities, ensuring that cross-modal interactions are still captured. This clarification addresses the concern to some extent but does not fully resolve the potential trade-off between efficiency and interaction modeling. While they claim this approach prevents negative interference, they do not provide direct empirical evidence comparing cross-modal interaction effectiveness between their sequential training and traditional joint optimization. Referring to Table 17 in the appendix is helpful but could be made more convincing by including detailed performance metrics that specifically measure the quality of cross-modal interactions (e.g., ablation studies focusing on tasks highly dependent on multimodal fusion).
> > > >
> > > > For Q5, the authors describe how parameter-efficient updates like LoRA provide implicit regularization and enhance model generalization. They back this with theoretical and empirical evidence from existing literature and articulate how their design draws inspiration from sparse Mixture-of-Experts (MoE) architectures. While the connection between their Modality-Specific Multi-Query Adapter (MMQA), self-gated multimodal query fusion, and generalization is reasonable, the explanation could be strengthened with specific experimental results demonstrating these effects. For instance, comparisons of CREMA’s generalization across unseen modalities or domains versus other frameworks would bolster their claims. The references to recent work on LoRA and MoE are apt, but their relevance would be more convincing if directly linked to empirical findings within the CREMA framework.

---

> > > > > ### Author Response · Authors · 2024-11-23
> > > > > **Response to Follow-up Questions on Q4/Q5**
> > > > >
> > > > > > For Q4, the authors explain that while weights for each modality are updated sequentially during training, the final loss computation includes all modalities, ensuring that cross-modal interactions are still captured. This clarification addresses the concern to some extent but does not fully resolve the potential trade-off between efficiency and interaction modeling. While they claim this approach prevents negative interference, they do not provide direct empirical evidence comparing cross-modal interaction effectiveness between their sequential training and traditional joint optimization. Referring to Table 17 in the appendix is helpful but could be made more convincing by including detailed performance metrics that specifically measure the quality of cross-modal interactions (e.g., ablation studies focusing on tasks highly dependent on multimodal fusion).
> > > > >
> > > > > Thank you for your constructive feedback.
> > > > >
> > > > > To further validate the cross-modal interaction effectiveness between sequential training and joint training in CREMA, we performed additional experiments as an extension of **Table 9**.
> > > > >
> > > > > Specifically, we compare the Hard accuracy of CREMA on SQA3D and NEXT-QA under two different training settings: sequential and joint. Hard accuracy indicates performance on the hard subset, where samples are selected if CREMA with only V fails to predict correctly in a zero-shot manner (i.e., the subset of 0% zero-shot accuracy of CREMA with V) , as described in Lines 483–484 of our submission. This means that input examples in the hard subset may require additional knowledge to find appropriate answers.
> > > > >
> > > > > As shown in the tables below, the fine-tuned performance of CREMA with modality sequential training surpasses that of CREMA with joint training by a significant margin. This demonstrates that the proposed modality sequential training is able to **interact more effectively with other modalities** during optimization and learn beneficial multimodal information to predict the hard subsets.
> > > > >
> > > > > Although CREMA with joint training also achieves significant performance improvement on the hard subset compared to zero-shot performance with V only, it results in lower performance compared to the sequential training setting. This suggests that, despite direct cross-modal interaction through joint optimization of multiple modalities, it struggles with negative interference when optimizing significantly distinct modalities simultaneously.
> > > > >
> > > > > **SQA3D**
> > > > > Model | Modalities | Training | Hard Acc.
> > > > > |-|-|-|-|
> > > > > CREMA |  V, P, D| Joint | 39.0
> > > > > CREMA |  V, P, D| Sequential | 42.1 (+3.1%p)
> > > > >
> > > > >
> > > > > **NEXT-QA**
> > > > > Model | Modalities | Training | Hard Acc.
> > > > > |-|-|-|-|
> > > > > CREMA |  V, F, D, N | Joint | 48.1
> > > > > CREMA |  V, F, D, N | Sequential | 50.0 (+1.9%p)
> > > > >
> > > > > ---
> > > > >
> > > > > > For Q5, the authors describe how parameter-efficient updates like LoRA provide implicit regularization and enhance model generalization. They back this with theoretical and empirical evidence from existing literature and articulate how their design draws inspiration from sparse Mixture-of-Experts (MoE) architectures. While the connection between their Modality-Specific Multi-Query Adapter (MMQA), self-gated multimodal query fusion, and generalization is reasonable, the explanation could be strengthened with specific experimental results demonstrating these effects. For instance, comparisons of CREMA’s generalization across unseen modalities or domains versus other frameworks would bolster their claims. The references to recent work on LoRA and MoE are apt, but their relevance would be more convincing if directly linked to empirical findings within the CREMA framework.
> > > > >
> > > > > We would kindly remind the reviewer that we have already included quantitative analyses demonstrating the generalizability of CREMA compared to other baselines across two key dimensions:
> > > > >
> > > > > - **Fine-Tuning Performance**: CREMA was evaluated across seven video-language reasoning tasks spanning eight distinct modalities: video, depth map, optical flow, surface normals, audio, thermal heatmap, touch map, and 3D point cloud. These experiments were conducted using two different backbones, BLIP-2 and Mistral-7B (**Table 6**), highlighting the generalizability and robustness of CREMA across diverse multimodal tasks.
> > > > > - **Zero-Shot Evaluations** (**Tables 5 and 16**): CREMA's generalization was further validated through zero-shot evaluations conducted on SQA3D (video + point cloud) and MUSIC-AVQA (video + audio), where it effectively handled unseen tasks without additional fine-tuning.
> > > > >
> > > > > These results clarify CREMA's capability to generalize effectively across both fine-tuned and zero-shot settings, providing strong empirical support for our claims. We appreciate the reviewer’s suggestion and are open to further clarifying these results if needed.

---

> > > > > > ### Comment · Reviewer_wvMi · 2024-11-23
> > > > > >
> > > > > > Thank you for addressing my concerns with additional experiments and detailed explanations.
> > > > > >
> > > > > > For Q4, I appreciate the extended analysis comparing sequential and joint training. The results on SQA3D and NEXT-QA clearly demonstrate the advantages of sequential training in effectively capturing multimodal interactions while mitigating negative interference. I am satisfied with your response.
> > > > > >
> > > > > > For Q5, The fine-tuning and zero-shot evaluation results across diverse tasks and modalities provide strong empirical support for CREMA’s generalization capabilities. Your explanation regarding the framework’s adaptability and robustness is clear. I am satisfied with your response to this question as well.

---

> > > ### Comment · Reviewer_wvMi · 2024-11-22
> > >
> > > Thank you for addressing my concerns regarding the use of a fully connected (FC) layer and the interpolation of video queries in your framework. I find your explanation and the accompanying empirical evidence satisfactory, which clearly demonstrates the advantages of your design choice. I am satisfied with your answer to Q2.
> > >
> > > I am also satisfied with your response to Q3. Your rationale for prioritizing video queries, supported by task-specific reasoning and experimental results, is clear. The additional experiments comparing different prioritization strategies further validate your design choice.

---

> ### Comment · Reviewer_wvMi · 2024-11-22
>
> Thank you for providing the results of your ablation study and addressing my concern regarding the impact of query token lengths on performance, trainable parameters, and computational cost. I am mostly satisfied with the answer to Q1. However, your claim that the Q-Former “removes irrelevant information” remains qualitative. I would appreciate it if you could provide further evidence or discussion to address this question.

---

> > ### Author Response · Authors · 2024-11-23
> > **Response to Follow-up Questions on Q1**
> >
> > > Thank you for providing the results of your ablation study and addressing my concern regarding the impact of query token lengths on performance, trainable parameters, and computational cost. I am mostly satisfied with the answer to Q1. However, your claim that the Q-Former “removes irrelevant information” remains qualitative. I would appreciate it if you could provide further evidence or discussion to address this question.
> >
> >
> > Thank you for your valuable feedback! We’re glad to hear that you are mostly satisfied with our response to Q1. To address your remaining concern about our claim regarding the Q-Former, we’d like to provide further clarification.
> >
> > Our statement that "Q-Former removes irrelevant information" is based on the original BLIP-2 paper [1], which describes the Q-Former as follows:
> >
> > ```
> > Q-Former is a lightweight transformer that employs a set of learnable query vectors to extract visual features from the frozen image encoder. It acts as an information bottleneck between the frozen image encoder and the frozen LLM, feeding the most useful visual features for the LLM to output the desired text.
> > ```
> >
> > From the authors’ perspective, the Q-Former serves as a compression module, distilling raw visual features (e.g., CLIP-Image features) into a smaller set of query tokens. This process prioritizes high-level semantic information (e.g., holistic scene understanding, object relationships) while potentially discarding finer details (e.g., precise object coordinates), as supported by prior analyses [2] (feel free to check interesting observations in this paper **Section 3.2**).
> >
> > While theoretically concatenating all visual tokens without compression could provide the LLM with more raw information, video inputs typically consist of multiple frames, making it crucial to balance token length and compression. **The Q-Former’s ability to compress tokens ensures efficient processing while preventing the LLM’s context window from being overwhelmed**, striking an important trade-off between maintaining essential information and handling longer sequences effectively.
> >
> >
> > Furthermore, quantifying "irrelevant" versus "relevant" information is inherently task-dependent, as it varies based on specific text queries or downstream tasks. To refine our claim, we now describe the Q-Former as:
> >
> > ```
> > This design enables the Q-Former to compress image tokens into a fixed-length set of query tokens, facilitating efficient processing of video inputs while preserving critical high-level information.
> > ```
> >
> > We have updated this claim in **Lines 180–183** and appreciate your insightful suggestion/question. Please let us know if further clarification/discussion is needed.
> >
> > [1] Blip-2: Bootstrapping language-image pre-training with frozen image encoders and large language models. ICML2023.
> > [2] DeCo: Decoupling Token Compression from Semantic Abstraction in Multimodal Large Language Models. Arxiv 2405.20985

---

> > > ### Comment · Reviewer_wvMi · 2024-11-23
> > >
> > > Thank you for the detailed clarification and the additional references to support your claim regarding the Q-Former. The updated description in your submission strikes a more precise tone by emphasizing the preservation of high-level semantic information, which aligns well with task-specific requirements. I am happy with the thoughtful refinement and the additional context you provided, which clarified the trade-offs involved in the Q-Former’s design.
> > >
> > > I am satisfied with your answer to Q1.

---

> ### Author Response · Authors · 2024-11-23
> **Thanks for your constructive feedback and discussion!**
>
> Dear Reviewer wvMi:
>
>   We appreciate your constructive feedback and insightful questions to strengthen our submission! We are glad to hear that you are satisfied with our rebuttal on your Q1/Q2/Q3/Q6.
>
>    We’ve also incorporated these additional discussions and experiments (e.g., query token length, modality prioritization, MLP for projection) in revision **Section B.8** based on your valuable feedback, which improved our paper. Thank you for your insightful comments!
>
> We also provide further discussions/clarifications about Q1/Q4/Q5 to make them more clear. We believe these updates would further address your concerns and make the paper more solid.
>
> If possible, we kindly request the reviewer to reconsider the score/rating. And please let us know if further clarifications or experiments are needed—we’re happy to provide them!
>
> Best,
>
> Authors

---

> > ### Comment · Reviewer_wvMi · 2024-11-23
> >
> > Dear Authors,
> >
> > Thank you for your detailed rebuttal and the additional discussions and experiments addressing my questions. I appreciate the effort you have put into refining your paper based on the feedback provided.
> >
> > I am satisfied with the revisions and clarifications provided for my questions, and I believe you have comprehensively addressed my concerns. The updates significantly strengthen the quality and clarity of the paper. Based on these improvements, I will increase the score to 8. Thank you again for your thoughtful and thorough responses.
> >
> > Best regards,
> >
> > Reviewer wvMi

---

> > > ### Author Response · Authors · 2024-11-24
> > > **Thank you for raising score!**
> > >
> > > Dear Reviewer wvMi,
> > >
> > > Thank you for taking the time to review and discuss our responses and revisions in detail. We are grateful for these thoughtful feedback and discussions.
> > >
> > > We truly appreciate your support and the updated score!
> > >
> > > Best,
> > >
> > > Authors

---

### Official Review · Reviewer_5ndk · 2024-10-28

**Soundness:** 3
**Presentation:** 3
**Contribution:** 3
**Rating:** 6
**Confidence:** 4

**Summary:**

This paper proposes a method, "CREMA," that addresses the problem of video understanding with diverse modalities, including optical flow, point clouds, audio, etc. CREMA first uses modality-specific encoders to encode each modality. Then CREMA introduces a Q-former to extract features from each modality. Before feeding the features into LLMs, CREMA further leverages a self-gating modality fusion guided by the video features. Such an approach has the advantage of significantly less trainable parameters and competitive performance across multiple datasets, including MUSIC-AVQA, SQA3D, etc.

**Strengths:**

* S1: The presentation of this paper is straightforward and clear.

* S2: The proposed fusion approach with Q-former (architecture) and modality-sequential training (training recipe) are both reasonable and looks simple for other researchers to follow.

* S3: The evaluation covers various domains, including audio, point clouds, optical flows, etc. The approach CREMA has demonstrated competitive performance across these scenarios, especially when the number of modalities is large.

**Weaknesses:**

* W1: This paper lacks sufficient quantitative or qualitative analysis on why multi-modality assists the model. For example, the MUSIC-AVQA performance in Table 1 can benefit from depth and surface normal information, which is not very intuitive. Therefore, some visualizations or other formats of analysis the authors see fit will greatly enhance the motivation here. I saw Figure 3 and the analysis provided by the authors. However, it is unclear whether the learned Q-former indeed considers these modalities, as indicated by Sec. B.7. Since the author uses self-gate to fuse the modalities, is it possible to analyze the model's reliance on certain modalities with attention scores?

* W2: Following the previous point, the increased number of trainable parameters with more modalities makes it complicated to confirm whether the additional modalities are indeed helpful. For example, adding depth and normal information increases the trainable parameters from 9M to 38M.

**Questions:**

* Q1: Why CREMA is called a video-language model? For example, SQA3D mainly uses RGBD as input, and the authors call CREMA a video-language model because the visual information is formatted as a video sequence.

* Q2: Although the authors have compared the trainable parameter number, it is arguable what is the number of total parameters, as LORA is used. The questions is: what is the total number of parameters, and what is the speed of inference?

* Q3: It is interesting to see that modalities of depth or surface normal are used, or even helpful, for MUSIC-AVQA and NExT-QA. I suggest the authors provide analysis or visualizations of how such modalities benefit the models.

---

> ### Author Response · Authors · 2024-11-20
> **Official Comment by Authors (1)**
>
> Thank you for your positive review and constructive comments. During the rebuttal period, we have made every effort to address your concerns. The detailed responses are below:
>
> > **W1**: This paper lacks sufficient quantitative or qualitative analysis on why multi-modality assists the model. For example, the MUSIC-AVQA performance in Table 1 can benefit from depth and surface normal information, which is not very intuitive. Therefore, some visualizations or other formats of analysis the authors see fit will greatly enhance the motivation here. I saw Figure 3 and the analysis provided by the authors. However, it is unclear whether the learned Q-former indeed considers these modalities, as indicated by Sec. B.7. Since the author uses self-gate to fuse the modalities, is it possible to analyze the model's reliance on certain modalities with attention scores?
>
> Thanks for your feedback on more concrete visualizations would help. We are trying to plot the attention map in the cross-attention layers in Q-former during this rebuttal. We are still implementing code for visualization and will update the visualization and analysis in the next few days.
>
> And we kindly remind that beyond Figures 3 and Section B.3, we also provide more analysis about:
> - **(Line 457-472) RQ 2**: How does CREMA address challenges and help video reasoning with more modalities?
> - **(Line 503-517) RQ 4**: The impact of new modalities on easy/hard questions.
>
> to future explain why those new modalities could help.
>
> ---
>
> > **W2**: Following the previous point, the increased number of trainable parameters with more modalities makes it complicated to confirm whether the additional modalities are indeed helpful. For example, adding depth and normal information increases the trainable parameters from 9M to 38M.
>
> Thank you for raising this point about the relationship between trainable parameters and the utility of additional modalities. To address this concern, we re-present a detailed comparison between CREMA and baseline methods, including BLIP-2 and 3D-LLM, across varying numbers of modalities and parameter sizes. Below are the results:
>
> Exp No. | Model (Modalities) | Trainable Param. | Acc.
> -|-|-|-|
> MUSIC-AVQA
> 1 | BLIP-2 (V) | 108M | 78.9
> 2 | CREMA (V) | 4M | 78.7
> 3 | BLIP-2 (A,V,F) | 324M | 78.1
> 4 | CREMA (A,V,F) | 21M | 80.5
> SQA3D
> 5 | 3D-LLM (V) | 108M | 51.8
> 6 | CREMA (V) | 4M | 51.4
> 7 | 3D-LLM (V,P,D,N) |  434M | 52.0
> 8 | CREMA (V,P,D,N) | 38M | 54.6
>
> As we listed in the above table (copied from parts of tables 1&2), we can observe that simply scaling trainable parameter size (BLIP-2/3D-LLM) could help performance only when we number of modalities are limited/small (Exp. 1&2, 5&6). **When we are facing more modalities, simply scaling trainable parameters to 300-400M failed to obtain performance gain** while our CREMA framework shows consistent gain along with more modality input.
>
> These results indicate that CREMA's performance gains are (also) due to the effective use of new modalities, not just more parameters. We hope this clarification makes it clear that adding modalities is indeed beneficial in our framework.
>
> ---
>
> > **Q1**: Why CREMA is called a video-language model? For example, SQA3D mainly uses RGBD as input, and the authors call CREMA a video-language model because the visual information is formatted as a video sequence.
>
> Thank you for your question. In principle, our CREMA framework design can be applied to any multimodal large language model, but we only focus on video-language reasoning tasks as a specific research domain. However, we would like to note that the video-language domain often offers rich and challenging scenarios with an integration of other additional valuable modalities like audio/depth map/optical flow/thermal/… that we extensively investigated in the paper.
>
> In this case, CREMA is designed for video-language reasoning tasks and builds upon the SoTA design of the prior video QA/reasoning model [1], which adapts image-language models for video tasks. Furthermore, in Table 6, we demonstrate the effectiveness of CREMA (including its multimodal Q-Former, MMQA modules, and modality-sequential training) when integrated with the VideoChat2 backbone.
>
> Regarding SQA3D, it is a 3D question-answering dataset focused on indoor environments. It provides multiple keyframes for each room, which can be interpreted as a low-FPS video. This aligns well with CREMA’s design as a video-language reasoning framework.
>
> [1] Self-chained image-language model for video localization and question answering. NeurIPS 23.

---

> > ### Author Response · Authors · 2024-11-20
> > **Official Comment by Authors (2)**
> >
> > > **Q2**: Although the authors have compared the trainable parameter number, it is arguable what is the number of total parameters, as LORA is used. The questions is: what is the total number of parameters, and what is the speed of inference?
> >
> > Thank you for your question. To clarify:
> >
> > - **Total number of parameters**: As shown in Table 5, the total number of parameters in CREMA is varied and ranges from **4.1B to 4.2B**, depending on the specific multimodal encoders included for a given downstream task.
> > For example, if it is the CREMA (V, A), it includes a visual encoder ($\sim$ 1B), audio encoder ($\sim$ 0.08B), Multimodal QFormer (including LoRA) + other FC layers ($\sim$ 0.1B), and LLM ($\sim$ 3B).
> > - **Speed of inference**: On a single NVIDIA A6000 GPU, CREMA achieves an inference speed of approximately **1.9 seconds** per example.
> >
> > > **Q3**: It is interesting to see that modalities of depth or surface normal are used, or even helpful, for MUSIC-AVQA and NExT-QA. I suggest the authors provide analysis or visualizations of how such modalities benefit the models.
> >
> > Thanks for your suggestion on more visualization on modalities, we have provided the answer to this question in W1.

---

> > > ### Comment · Reviewer_5ndk · 2024-11-22
> > >
> > > I am grateful to the authors for the rebuttals, and my concerns on parameter counts and fair comparisons are addressed. However, I still have some follow-up questions:
> > >
> > > * I checked the RQ2 and RQ4 suggested by the authors. But it is still mysterious why modalities like depth and surface normal are helpful for video reasoning. Could the authors provide any intuitions?
> > >
> > > * > we can observe that simply scaling trainable parameter size (BLIP-2/3D-LLM) could help performance only when we number of modalities are limited/small (Exp. 1&2, 5&6)
> > >
> > > Just out of curiosity, do the authors have any hypotheses on this? This is a very interesting observation.

---

> > > > ### Author Response · Authors · 2024-11-23
> > > > **Response to Reviewer 5ndk**
> > > >
> > > > Dear Reviewer 5ndk,
> > > >
> > > > Thank you for your thoughtful follow-up questions. We are glad that our rebuttal addressed your initial concerns regarding parameter counts and fair comparisons.
> > > >
> > > > Below, we provide additional explanations and insights based on your queries:
> > > >
> > > > > I checked the RQ2 and RQ4 suggested by the authors. But it is still mysterious why modalities like depth and surface normal are helpful for video reasoning. Could the authors provide any intuitions?
> > > >
> > > >
> > > > FYI, we’ve added a new qualitative visualization of model attention maps in **Appendix Figure 5**. These maps reveal that without optical flow input, the model's attention becomes diffuse and unfocused, while with optical flow, it concentrates on dynamic regions  (area with motion), improving performance.
> > > >
> > > > We believe this is due to optical flow helping the model identify which part of the video remains static, aiding in deducing that a sound likely doesn’t originate from a static middle instrument.
> > > >
> > > > Similarly, other modalities provide information that videos alone lack. Depth maps contribute useful **spatial cues** for questions like ```“Is the clock nearer than the sofa?”```. Surface normals add **shape information**, aiding questions such as ```“What material is this object made of?”``` These diverse modalities enrich the model’s understanding and reasoning capabilities.
> > > >
> > > > ---
> > > > > we can observe that simply scaling trainable parameter size (BLIP-2/3D-LLM) could help performance only when we number of modalities are limited/small (Exp. 1&2, 5&6). Just out of curiosity, do the authors have any hypotheses on this? This is a very interesting observation.
> > > >
> > > > This is indeed an interesting observation. We kindly remind that our experiments primarily focus on fine-tuning models on downstream datasets with limited samples (e.g., ~34K in NExT-QA, ~26K in SQA3D, and ~31K in MUSIC-AVQA).
> > > >
> > > > We hypothesize that larger trainable parameters overfit more easily in such limited data scenarios, reducing their effectiveness. In contrast, our modality-specific LoRA design mitigates overfitting by enabling efficient parameter usage, aligning with prior findings [1,2] that LoRA outperforms full fine-tuning in data-constrained tasks.
> > > >
> > > > Additionally, We further emphasize that certain modalities (e.g., thermal/tactile maps in Tabel 4) are rare and costly to collect at scale. Our CREMA offers a cost-effective, efficient approach to quickly adapt to these rare modalities while maintaining strong performance. We deeply appreciate your insightful observations and constructive feedback. These have significantly strengthened our paper.
> > > >
> > > > [1] LoRA Learns Less and Forgets Less. Arxiv 2405.09673.
> > > > [2] LoRA vs Full Fine-tuning: An Illusion of Equivalence. Arxiv 2410.21228.
> > > >
> > > > ---
> > > >
> > > > We hope the added clarifications and visualizations address your concerns and kindly request to further reconsider the rating/scoring. We are happy to provide further details or results if needed. Thank you for your time and valuable input!

---

> > > > > ### Comment · Reviewer_5ndk · 2024-11-23
> > > > >
> > > > > Thank you for the additional clarifications! I don't have further questions.

---

> > > > > > ### Author Response · Authors · 2024-11-27
> > > > > > **Thanks for your response**
> > > > > >
> > > > > > Dear Reviewer 5ndk,
> > > > > >
> > > > > > Thank you for taking the time to review our responses and for acknowledging our clarifications. We appreciate your thoughtful feedback and are pleased to hear there are no remaining questions.
> > > > > >
> > > > > > If there are any partially addressed issues or additional areas where we could further improve the paper, we would be grateful for your guidance on how we might improve the paper to the point where it would earn a clear "Accept" from you.
> > > > > >
> > > > > > Thank you again for your time and constructive input.
> > > > > >
> > > > > > Best regards,
> > > > > > Authors

---

> ### Comment · Reviewer_5ndk · 2024-12-02
>
> Dear Authors,
>
> Thank you again for your rebuttals! I regret to say that I would fix my ratings at 6. Here are my rationales:
>
> In general, the Q-former has been a natural solution for modality fusion, which decreases my "surprise" or "knowledge" when reading your paper, especially when your paper does not provide a precise intuition and generalizable analysis about "why the additional modalities help?" (Good examples for this would be the "Vision Transformers Need Registers" and "Frozen Transformers in Language Models are Effective Visual Encoder Layers" from last year's ICLR.) However, your exploration of Q-former and extensive evaluation, especially with less trainable parameters and higher performance, is meaningful. So my rating is finally 6.
>
> Everyone has their standard for rating a paper. My principle is that I would give all the qualified papers a six (your paper is definitely in this category), help them get accepted, and give an eight to the papers with clear physical intuitions/explanations, which I really like.
>
> I hope the above reviewer's response addresses the authors' concerns as a rebuttal.
>
> Best,
> Reviewer

---

> > ### Author Response · Authors · 2024-12-03
> >
> > Dear Reviewer 5ndk,
> >
> > Thank you for your response and for taking the time to provide detailed feedback.
> >
> > We completely understand your rationale and respect your decision. Thanks for pointing out good examples that we can learn more about for future work.
> >
> > And thanks for your support and positive rating again!
> >
> > Best,
> >
> > Authors

---

### Official Review · Reviewer_dN8u · 2024-11-02

**Soundness:** 3
**Presentation:** 3
**Contribution:** 3
**Rating:** 8
**Confidence:** 4

**Summary:**

This paper proposes CREMA, a generalizable and modular modality-fusion framework that augments multiple modalities without extra human annotation and incorporates them into a query transformer, enhancing video reasoning. It introduces a progressive multimodal fusion design, maintaining computational efficiency and improving performance. Validated on 7 video-language reasoning tasks, CREMA outperforms or matches strong multimodal LLMs while significantly reducing trainable parameters, demonstrating its effectiveness and innovation.

**Strengths:**

1. The paper is clear writing and easy to follow.
2. Few current works focus on integrating multiple modalities, so the authors' motivation is commendable.
3. I appreciate the paper's innovation. Although it may not introduce many new structures, the modality-adaptive early exit strategy appears to have broad application potential. It's the first time I've seen the use of gradients to determine whether to exit early, and it is also the first method to apply early stopping by modality. Therefore, I acknowledge the paper's innovative approach.

**Weaknesses:**

1. Overall, I believe this paper is worthy of acceptance and presents no significant issues. My only curiosity, as mentioned by the authors in the limitations section, is whether the method can be applied to more advanced baselines such as LLava, rather than just BLIP. If feasible, I would appreciate the authors addressing this point, which could lead me to adjust my score upwards.

**Questions:**

Please refer to the weakness

---

> ### Author Response · Authors · 2024-11-20
> **Official Comment by Authors (1)**
>
> Thank you for your positive feedback and for recognizing the unique novel of our CREMA framework. During the rebuttal period, we have made every effort to address your concerns. The detailed responses are below:
>
> > **W1**: Overall, I believe this paper is worthy of acceptance and presents no significant issues. My only curiosity, as mentioned by the authors in the limitations section, is whether the method can be applied to more advanced baselines such as LLava, rather than just BLIP. If feasible, I would appreciate the authors addressing this point, which could lead me to adjust my score upwards.
>
> Thank you for your positive feedback and for raising this important point. We agree that applying our method to more advanced baselines is valuable.
>
> In fact, **we have already applied CREMA to stronger LLM backbones beyond BLIP2**. In **Table 6** and at the end of **Section 4.2**, we report experiments where we integrated CREMA with VideoChat2 using the Mistral-7B backbone. We observed consistent performance gains when incorporating additional modalities, while keeping the number of trainable parameters relatively small. Here is a part of the copied results from Table 6 as a quick view:
>
> Model (Modality) |  LLM | NExT-QA-Acc.
> |-|-|-|
> Video-LLaMA (V) | Vicuna-7B | 60.6 |
> LLaVA-NeXT (V)  | Qwen1.5-7B | 78.2
> VideoChat2 (V)  | Mistral-7B | 78.4 |
> CREMA (V, F)  | Mistral-7B | 78.9 |
> CREMA (V,F,D) | Mistral-7B | **79.4** |
>
> Our CREMA-Mistral-7B model achieves the best performance among these strong video-language models with similar LLM sizes, demonstrating the effectiveness of our method when applied to advanced backbones.
>
> In this rebuttal, we also conducted new experiments applying CREMA to Video-LLaVA with the Vicuna-7B backbone following your suggestion. We observed similar improvements/effectiveness of the CREMA framework.
>
> Method        |       LLM      | NExTQA-Acc.
> |-|-|-|
> Video-LLaVA (V)  | Vicuna-7B | 66.3
> CREMA (V, F, D)   | Vicuna-7B | **67.9**
>
> These results show that our CREMA framework **consistently enhances performance** with more modalities across different vision-language backbones, including LLaVA. We appreciate your suggestion and hope this addresses your concern and demonstrates the applicability of CREMA to more advanced models. Thank you again.

---

> ### Author Response · Authors · 2024-11-25
> **Thanks for your reviewing and a gentle reminder**
>
> Dear Reviewer dN8u,
>
>
> Thank you for your time and effort in reviewing our paper. We kindly notify you that the end of the discussion stage is approaching. Could you please check if your concerns/questions are addressed in our rebuttal? During the rebuttal period:
>
> - we provide the results of the CREMA framework with other MLLM (VideoChat2, VideoLLaVa).
>
> We hope the added clarifications and the revised submission address your concerns and kindly request the review to further reconsider the rating/scoring if possible. We are happy to provide further details or results if needed.
>
>
> Best,
>
> Authors

---

> ### Comment · Area_Chair_vpQf · 2024-11-27
>
> Dear reviewer,
>
> Today is the last day for reviewers to ask questions to authors. Did the authors' rebuttal address your concern? Do you have any additional questions?

---

> ### Comment · Reviewer_dN8u · 2024-11-27
>
> Thank you for the response. The experimental results from the authors are very meaningful, and I have decided to raise the score to 8.

---

> > ### Author Response · Authors · 2024-11-27
> >
> > Thanks for your engagement and for increasing your score. We are glad we adequately addressed your concerns.
> >
> > Best regards,
> > Authors

---

### Official Review · Reviewer_d9Rm · 2024-11-06

**Soundness:** 3
**Presentation:** 2
**Contribution:** 2
**Rating:** 6
**Confidence:** 4

**Summary:**

The paper introduces CREMA, an efficient and generalizable framework for video-language reasoning that enhances understanding through multiple modalities, including video, depth, audio, and 3D point cloud data, among others. CREMA employs a modular fusion approach, with lightweight, modality-adaptive modules that allow for easy integration of new modalities with minimal added parameters. The framework also incorporates a novel self-gated attention fusion technique to reduce computational demands. Additionally, it proposes a modality-sequential modular training and adaptive early exit strategy to boost training efficiency and enable faster adaptation to new modalities. CREMA demonstrates superior performance across multiple benchmarks, such as SQA3D, MusicQA, NExT-QA, TouchQA, and ThermalQA, highlighting the benefits of integrating diverse input modalities for improved video reasoning.

**Strengths:**

- This paper proposes a framework capable of handling multiple modalities and addresses the issue of token quantity increasing with the number of modalities.

- A single Q-former is used to process multiple modalities, avoiding the large increase in parameters typically associated with multi-modal input. Each modality requires only a small amount of modality-specific parameters, and since the parameters for each modality within the Q-former are independent, processing different modalities does not cause interference.

- The modality-sequential and modular training approach accommodates the differences across various modalities, preventing overfitting or underfitting to any specific modality.

- The paper demonstrates through multiple benchmarks that the proposed framework effectively integrates information from diverse modalities, thereby enhancing video reasoning capabilities.

**Weaknesses:**

- I believe the main focus of this paper is on ensuring that the number of tokens input into the LLM does not increase linearly with the number of modalities, while maximizing parameter sharing across modalities to avoid excessive parameter growth. However, I feel that the teaser image does not effectively highlight these key points.

- My biggest concern lies with the Self-gated Multimodal Query Fusion. This module concatenates tokens from different modalities along the channel dimension, meaning that the input modalities during inference must match those used in training exactly—neither more nor less—otherwise, there will be a parameter mismatch within the Self-gated Multimodal Query Fusion. Many videos, for example, may not contain point cloud information; however, if point cloud data was included as input during training, it must also be part of the input during inference. This limitation significantly restricts the flexibility of input modality types.

- Additionally, the description of the zero-shot setup is not clear enough. Before performing zero-shot evaluation on SQA3D and MUSIC-AVQA, which datasets were used to train and optimize the model's new parameters? Furthermore, as mentioned above, I believe that the Self-gated Multimodal Query Fusion limits the model's zero-shot reasoning capabilities, as different combinations of input modalities would require different models. This implies that different models were likely used for zero-shot evaluation on SQA3D and MUSIC-AVQA. Therefore, the authors should clarify which specific model was used for evaluation in each experiment.

- Some related works on integrating multiple modalities are missing, such as MultiPLY[1] and X-VILA[2], both of which are multimodal LLMs capable of handling various input modalities. The authors should discuss the relationship with these works.

[1]. MultiPLY: A Multisensory Object-Centric Embodied Large Language Model in 3D World
Yining Hong, Zishuo Zheng, Peihao Chen, Yian Wang, Junyan Li, Chuang Gan

[2]. X-VILA: Cross-Modality Alignment for Large Language Model
Hanrong Ye, De-An Huang, Yao Lu, Zhiding Yu, Wei Ping, Andrew Tao, Jan Kautz, Song Han, Dan Xu, Pavlo Molchanov, Hongxu Yin


---

Post-rebuttal:

Most of concerns are addressed, I raise my rating to 6.

**Questions:**

Please refer to the weakness section.

---

> ### Author Response · Authors · 2024-11-20
> **Official Comment by Authors (1)**
>
> Thanks for the valuable comments. In this rebuttal, we have made every effort to address your concerns. The detailed responses are below:
>
> > **W1**: I believe the main focus of this paper is on ensuring that the number of tokens input into the LLM does not increase linearly with the number of modalities, while maximizing parameter sharing across modalities to avoid excessive parameter growth. However, I feel that the teaser image does not effectively highlight these key points.
>
> We respectfully re-emphasize that CREMA does not only focus on minimizing token and parameter growth, but also aims to **effectively** leverage diverse multimodal information that other baseline methods fail to handle. Those are clearly explained in:
>
> **Lines 96-101**: “...some modalities may be redundant or irrelevant to the reasoning tasks, and optimizing with all modalities simultaneously can lead to a certain deficiency…”
>
> **Lines 128-129**: “We show the efficacy of CREMA on seven video reasoning datasets by achieving better/equivalent performance…”
>
> Our experiments show that other baseline methods (with the same multimodal input) struggle with more modalities—even they are with linearly increased tokens and require much more memory (**Table 15**), computation (**Table 1-3**), and training time (**Table 7**) but achieve lower performance.
>
> Without our multimodal modular fusion and modality-sequential training design, those baseline methods face challenges of many modality optimizations in terms of both **effectiveness** and **efficiency**.
>
> In contrast, CREMA provides a lightweight solution that requires fewer resources, avoids modality interference during optimization, and achieves better results, especially with multiple modalities. This demonstrates that CREMA is both efficient and effective in handling diverse modalities.
>
> We appreciate your suggestion about the teaser image and **have updated it (Figure 1)** in our revision to better highlight these key points.
>
> ---
> > **W2**: Concerns regarding Self-gated Multimodal Query Fusion and missing modality during inference. Many videos, for example, may not contain point cloud information; however, if point cloud data was included as input during training, it must also be part of the input during inference. This limitation significantly restricts the flexibility of input modality types.
>
> Thank you for your insightful feedback regarding the flexibility for missing modalities. In our current setup, we maintain the same combination of modalities during both training & inference without missing any modalities. We acknowledge that in real-world scenarios, some modalities might be unavailable during inference, which could restrict the flexibility of our framework.
>
> However, it is important to emphasize that, to the best of our knowledge, **no existing MLLM baselines** are capable of such general-purpose intrgration of a variety of modalities while effectively addressing the missing modality issue.   While the current implementation of CREMA does not explicitly address missing modality issues, its modularized framework and modality sequential training   and the proposed automatic early exit mechanisms provide a strong foundation for exploring such capabilities in the future.
>
>
> To partially clarify this concern, in this rebuttal, we experimented with a simple solution within the existing CREMA framework. When certain modalities are missing at inference time, we **skip the fusion layer** and **directly concatenate** all available multimodal tokens before feeding them into the LLM, so it simply avoids parameter mismatches.
>
> |Exp No.|	Setting (Input Modalities)|	SQA3D Acc. (%)|
> |-|-|-|
> |1|No Missing (V, P, D)|53.1|
> |2|Drop D (V, P)|51.1|
> |3|Drop D and P (V)|49.9|
>
> |Exp No.|	Setting (Input Modalities)|SQA3D Acc. (%)|
> |-|-|-|
> 4|No Missing (V, P)|52.1|
> 5|Drop D from Setting 1 (V, P)	| 51.1
> 6|No Missing (V)|51.8
> 7|Drop D and P from Setting 1 (V)|49.9
>
> Note: V = Video, P = Point Cloud, D = Depth.
>
> Here, "No Missing" means the model was trained and evaluated with the same complete modality set. Comparing Settings 1&2, and 1&3, we observe that dropping modalities during inference leads to a decrease in performance (a 2% drop when dropping D, and an extra 1.2% when dropping P). However, the decrease is not drastic, indicating that the model remains effective even with missing modalities. And this decrease is also reasonable since the model is not optimized with the dropping condition during training.
>
> Furthermore, the comparisons between models well-trained with fewer modalities (Settings 4&6) and those where modalities were dropped during inference (Settings 2&3) show that CREMA exhibits robustness to missing modalities.
>
> While this simple method helps, we also agree that incorporating more advanced designs [1,2] to handle missing modalities could enhance flexibility. Exploring techniques for handling dynamic modality combinations is an interesting direction for our future work.
>
> (continued)

---

> ### Author Response · Authors · 2024-11-20
> **Official Comment by Authors (2)**
>
> To the best of our knowledge, CREMA is the first framework that can seamlessly and effectively combine new modalities to assist video-language reasoning. We believe our work takes a solid step toward providing a versatile model backbone for those interesting future works. Thank you again for your valuable feedback.
>
> [1] Multimodal prompting with missing modalities for visual recognition, CVPR 2023.
> [2] Multimodal Representation Learning by Alternating Unimodal Adaptation. CVPR 2024
>
> ---
>
>
> > **W3**: Additionally, the description of the zero-shot setup is not clear enough. Before performing zero-shot evaluation on SQA3D and MUSIC-AVQA, which datasets were used to train and optimize the model's new parameters? Furthermore, as mentioned above, I believe that the Self-gated Multimodal Query Fusion limits the model's zero-shot reasoning capabilities, as different combinations of input modalities would require different models. This implies that different models were likely used for zero-shot evaluation on SQA3D and MUSIC-AVQA. Therefore, the authors should clarify which specific model was used for evaluation in each experiment.
>
> We included extra training details for zero-shot setting in Appendix Section A.3 (Line 948-954), and here is more clarification:
> For the zero-shot evaluation, the CREMA framework was trained as follows:
>
> - MMQA-Audio: Trained on AudioCaps data
> - MMQA-3D: Trained on the 3D-LLM QA dataset
>
> During training, all other parts of the CREMA framework remained frozen, ensuring that only the modality-specific modules were optimized for their respective tasks.
>
> In the zero-shot setting, we conducted evaluations on SQA3D (video + point cloud) and MUSIC-AVQA (video + audio). Since these tests include only two modalities at a time, we **bypassed** the Self-Gated Multimodal Query Fusion module and directly concatenated the video tokens with the corresponding modality tokens. This ensures **no parameter mismatch or interference** and every modality is **independent** during inference.
>
> Thus, the same base model was used for all zero-shot experiments (Table 5), with the appropriate MMQA module activated for the corresponding modality (audio or 3D). No separate models were trained for different zero-shot evaluations. We appreciate your suggestion and **have revised the paper (Line 1011-1015)** to make this setup clearer. Thank you again for your constructive feedback.
>
> > **W4**: Some related works on integrating multiple modalities are missing, such as MultiPLY[1] and X-VILA[2], both of which are multimodal LLMs capable of handling various input modalities. The authors should discuss the relationship with these works.
>
> Thank you for pointing out these related works. We would like to clarify the differences between MultiPLY, X-VILA, and our CREMA framework:
>
> - MultiPLY is a multisensory embodied LLM designed for interaction within 3D environments using a fixed set of modalities. In contrast, CREMA focuses on adapting to new modalities to assist diverse video-language reasoning tasks. Our framework emphasizes modality extensibility and efficiency, allowing seamless integration of any additional modalities.
> - X-VILA is an omni-modality model aimed at cross-modality alignment, understanding, and generation. While X-VILA concentrates on large-scale cross-modality alignment and generative tasks, CREMA is dedicated to effectively and efficiently leveraging diverse multimodal information specifically for specific tasks.
>
> We **have included these discussions and comparisons in our revision (Line 155-157)** to highlight them. Thank you again for your valuable feedback.

---

> ### Author Response · Authors · 2024-11-25
> **Thanks for your reviewing and a gentle reminder**
>
> Dear Reviewer d9Rm,
>
> Thank you for your time and effort in reviewing our paper. We kindly notify you that the end of the discussion stage is approaching. Could you please check if your concerns/questions are addressed in our rebuttal? During the rebuttal period:
>
> - we updated **Figure 1** as the reviewer suggested.
> - we provided new results for CREMA with missing modality.
> - we added more clarification for zero-shot CREMA training/inference settings in **Line 1011-1015**.
> - we added comparisons with related work VILA and MultiPLY in **Line 155-157**.
>
> We hope the added clarifications and the revised submission address your concerns and kindly request the review to further reconsider the rating/scoring, if possible. We are happy to provide further details or results if needed.
>
> Best,
>
> Authors

---

> ### Comment · Area_Chair_vpQf · 2024-11-27
>
> Dear reviewer,
>
> Today is the last day for reviewers to ask questions to authors. Did the authors' rebuttal address your concern? Do you have any additional questions?

---

> ### Author Response · Authors · 2024-12-01
> **We have less than two days left in the discussion period.**
>
> Dear Reviewer d9Rm,
>
> We sincerely appreciate your efforts in reviewing our paper and your constructive comments. Since there are less than two days left in the discussion period, could you please read our responses to check if your concerns are clearly addressed? We believe that our responses resolved all your concerns.
>
> We understand that the criteria for rating a paper can sometimes be subjective; however, if you agree that our work does not have remaining major concerns, we would like to kindly suggest your re-evaluation of the initial rating of this submission.
>
> Please let us know if you have any remaining questions, and we will be more than happy to address them.
>
> Best,
> Authors

---

### Author Response · Authors · 2024-11-20
**General Comments for Authors Rebuttal**

We thank the reviewers for their time and valuable comments. We appreciate that reviewers recognized:

- Commendable motivation of video-language + any modalities design (```dN8u```)
- Novelty of CREMA framework. (```dN8u```, ```wvMi```)
- Reasonable model and training stragtegy design (```d9Rm```, ```dN8u```, ```5ndk```)
- Strong potential for motivating future study (```5ndk```, ```dN8u```)
- Extensive experiments and strong results (```d9Rm```, ```dN8u```, ```5ndk```, ```wvMi```, ```bc5T```)
- Clear writing and paper flow (```dN8u```, ```5ndk```)

In the responses, we include more clarification and experiments as follows.
- Discussion on self-gated fusion (```d9Rm```)
- Comparison with X-VILA and MultiPLY (```d9Rm```)
- Clarification on zero-shot setting (```d9Rm```)
- Extra qualitative visualization (```5ndk```)
- Discussion on the effectiveness of more modalities (```5ndk```)
- Clarification on total parameters and running speed (```5ndk```)
- Clarification on modality sequential training (```wvMi```)
- Clarification on regularization effect (```wvMi```)
- Discussion on CREMA for remote sensing data (```wvMi```)
- Clarification on CREMA novelty (```bc5T```)
- Clarification on model comparison settings (```bc5T```)
- Exp1: missing modality (```d9Rm```)
- Exp2: CREMA with Video-LLaVa backbone (```dN8u```)
- Exp3: different number of query tokens (```wvMi```)
- Exp4: prioritizing different modalities (```wvMi```)
- Exp5: ablation on sigmoid function in self-gated fusion (```bc5T```)

We hope our replies can address the concerns, and please let us know if there are any new questions.

---

### Meta-Review · Area_Chair_vpQf · 2024-12-23

**Metareview:**

This paper was reviewed by 5 experts in the field. The authors' rebuttal resolved most of the concerns, and reviewers unanimously agreed to accept the paper.

The AC agrees with the reviewers' assessments and does not find strong reasons to overturn the reviewers' consensus. The decision is to recommend the paper for acceptance. The reviewers did raise some valuable suggestions in the discussion that should be incorporated in the final camera-ready version of the paper. The authors are encouraged to make the necessary changes to the best of their ability.

**Additional Comments On Reviewer Discussion:**

Most concerns were addressed during rebuttal, and reviewers unanimously agreed to accept the paper.

---

### Decision · Program_Chairs · 2025-01-22

Accept (Poster)